# Suitability of CICE Sea Ice Model for Seasonal Prediction and Positive Impact of CryoSat-2 Ice Thickness Initialization

Shan Sun[1] and Amy Solomon[2]

[1]NOAA Global Systems Laboratory, USA

[2] University of Colorado Boulder, Cooperative Institute for Research in Environmental Sciences and NOAA Physical Sciences Laboratory, USA

**Correspondence:** Shan.Sun@noaa.gov

**Abstract.**

The Los Alamos sea ice model (CICE) is being tested in standalone mode to identify biases that limit its suitability for seasonal prediction, where CICE is driven by atmospheric forcings from the NCEP Climate Forecast System Reanalysis (CFSR) and a built-in mixed layer ocean model in CICE. The initial conditions for the sea ice and mixed layer ocean are also from CFSR in the control experiments. The simulated sea ice extent agrees well with observations during the warm season at all lead times up to 12 months, in both the Arctic and Antarctic. This suggests that CICE is able to provide useful sea ice edge information for seasonal prediction. However, the model's Arctic sea ice thickness forecast has a positive bias that originates from the initial conditions. This bias often persists for more than a season, which limits the model's seasonal forecast skill. To address this limitation, additional CS2_IC experiments were conducted, where the Arctic ice thickness was initialized using CryoSat-2 satellite observations while keeping all other initial fields the same as in the control experiments. This reduced the positive bias in the ice thickness in the initial conditions, leading to improvements in both the simulated ice edge and thickness at the seasonal time scale. This indicates that CICE has the potential to improve its seasonal forecast skill and provide more accurate predictions of sea ice extent and thickness, when initialized with a more realistic sea ice thickness. This study highlights that the suitability of CICE for seasonal prediction depends on various factors, including initial conditions such as sea ice thickness, in addition to sea ice coverage, as well as oceanic and atmospheric conditions.

## 1 Introduction

Sea ice concentration observations from passive microwave satellites reveal that the Arctic has experienced a rapid decline in ice coverage over the past few decades, making it the region with the greatest warming on Earth (Chapman and Walsh, 2003; IPCC, 2014). Global climate models suggest that further decreases are expected in the coming years (e.g., IPCC, 2014, 2021). Sea ice plays a crucial role in the regional energy balance and the global climate as a whole. Acting as a thin material layer between the atmosphere and ocean, sea ice amplifies radiative feedback with its higher surface albedo compared to open water (e.g., Holland and Bitz, 2003; ACIA, 2005; Dethloff et al., 2006), making it a sensitive and visible indicator of climate change. Reliable sea ice prediction is essential not just for the polar regions but also for improving predictability in mid-latitudes at subseasonal to seasonal (S2S) time scales due to teleconnections (e.g., Randall et al., 1998; Jaiser et al., 2012; Li et al., 2014).

Identifying sources of predictability for weather at S2S time scales is challenging. However, sea ice has shown promise due to its seasonal anomaly persistence (Blanchard-Wrigglesworth et al., 2011a; Holland et al., 2011; Smith et al., 2016; Bushuk et al., 2019). Recent advances in sea ice modeling offers potential for improving both medium-range and climate predictions (e.g., Wang et al., 2013; Hebert et al., 2015; Guemas et al., 2016; Chevallier et al., 2017). Weather predictions from a numerical model incorporating a sea ice model are found to be more skillful compared to those based on Arctic sea ice persistence (Sigmond et al., 2013; Hebert et al., 2015). The accuracy of sea ice initial conditions is critical to seasonal sea ice forecast (Holland et al., 2011; Blanchard-Wrigglesworth et al., 2011b; Wang et al., 2013). In particular, the fidelity of sea ice thickness initialization is important as thickness has greater persistence than concentration (Krinner et al., 2010; Chevallier and Salas-Mélia, 2012; Day et al., 2014a; Collow et al., 2015; Allard et al., 2018; Blockley and Peterson, 2018). Furthermore, sea ice predictability is season-dependent (Holland et al., 2011; Day et al., 2014b; Bushuk et al., 2020) and the predictability of minimum sea ice extent depends on spring and early summer atmospheric conditions, as well as the inclusion of melt ponds (Liu et al., 2015; Schröder et al., 2019; Bushuk et al., 2020).

The Los Alamos Community Ice CodE (CICE; Hunke et al., 2015) has been selected to be integrated into NOAA's Unified Forecast System (UFS) as the next operational coupled atmosphere-ocean-sea ice-land system for S2S predictions. CICE, originally developed for long-term climate research, has been used in seasonal prediction applications with success and challenges, for example, in the Global Seasonal forecast systems at the UK Met Office and the Canadian Seasonal to Interannual Prediction System (CanSIPSv2) (Arribas et al., 2011; MacLachlan et al., 2015; Lin et al., 2020). Notably, Martin et al. (2023) attributed the enhanced skill in CanSIPSv2 compared to the previous version to an improved sea ice initialization procedure. In this study, we aim to assess the CICE's seasonal performance from a different perspective by examining it in standalone mode. This approach avoids various feedbacks associated with a fully coupled atmosphere-ocean-sea ice model, enabling a detailed analysis of sea ice conditions in this controlled environment. For instance, in standalone mode, Schröder et al. (2019) found the default conductivity coefficient in CICE to be too low at colder temperatures. The objective of this study is to identify first-order biases that limit the seasonal prediction skill of CICE and establish a consistent set of baseline sea ice forecasts for future studies that incorporate sea ice into coupled atmosphere and ocean modeling. Furthermore, the impact of biases in sea ice thickness initialization on forecast skill is investigated. The experimental setup is described in detail in Section 2. The basin-wide and regional performance at different lead times, as well as with different initializations, are presented in Section 3, followed by a summary and conclusion in Sections 4 and 5.

## 2 Model Setup

In this study, we utilized version 5.1.2 of the CICE model (Hunke et al., 2015), which is a dynamic-thermodynamic sea ice model. CICE simulates the growth, melting, and movement of sea ice in cold regions, incorporating a subgrid-scale ice thickness distribution. The model uses the Elastic-Viscous-Plastic (EVP) rheology for ice dynamics (Hunke, 2001), as well as the ice strength parameterization for the ridging scheme (Lipscomb et al., 2007). Our experiments adapted standard five ice thickness categories (a low bound thickness of 0., 0.64m, 1.39m, 2.47m and 4.57 m, respectively), with four ice layers and

one snow layer in each thickness category. A linear function of salinity for the freezing temperature was employed. A detailed description of the CICE model is available at Hunke et al. (2015).

The CICE experiments conducted in this study were performed in the global domain, utilizing a combination of an Arctic bipolar grid projection and a Mercator projection for the rest of the globe, as shown in Fig.1 of Bleck and Sun (2004). The experiments employed a horizontal resolution of 15 km at the North Pole and 30 km at 60°N and 60°S. The bathymetry data from ETOPO1 (Amante and Eakins, 2009) are interpolated onto the model's global compound grid. A time step of 900 seconds is used in all model experiments.

## 2.1 Atmospheric Boundary Conditions

The prescribed time-varying atmospheric boundary forcings used in this study are derived from the 6-hourly archives obtained from CFSR reanalysis[1] (Saha et al., 2010), which include downward surface radiation of both shortwave and longwave components, 10 m wind speed, 2 m temperature, 2 m specific humidity, and precipitation, which is further broken down into rain and snow. This CFSR atmospheric data covers the period from 2011 to 2017 and has a spatial resolution of 0.2°. A pre-processing step was undertaken to horizontally interpolate these variables onto the compound model grid.

## 2.2 Ocean Boundary Conditions

CICE requires information on the sea surface temperature (SST), which can be either prescribed directly or generated inline using the built-in mixed layer ocean model within CICE. Initially, the CFSR SST data was prescribed to drive the CICE model. However, it was found that this approach led to an unrealistically large basal melt, primarily attributed to a small yet persistent positive bias in the CFSR SST data, despite the success in Guemas et al. (2014), where ocean temperature and salinity are nudged towards the NEMOVAR ORAS4 ocean reanalysis (Mogensen et al., 2011). To address this issue and ensure consistency between the SST and the ice state, an alternative method was adopted, i.e., to use the mixed layer ocean model within CICE to prognose the SST. This means that the CICE model itself generates the SST information based on the physical processes happening within the mixed layer of the ocean. By employing this approach, the SST and ice state remain consistent, allowing for a more reliable representation of the interaction between the sea ice and the ocean in the model simulations than the earlier attempt. However, it's important to note that the mixed layer ocean model used in this approach is a simplified one-dimensional stationary model that does not include horizontal advection in the ocean. This limitation does impact the model results, as will be demonstrated later.

During the model integration, the temperature at the interface between the ice and ocean is maintained at the freezing temperature of the mixed layer. In case there is any excess energy remaining after the ice and snow have melted, it is transferred to the ocean mixed layer. The thickness of the mixed layer ocean model is held constant at 20 meters.

---

[1]available at http://rda.ucar.edu/datasets/ds094.0

**Table 1.** Details in two sets of CICE experiments.

| Experiments | Atm Boundary Conditions | Initial Conditions for Ice & Ocean | Initialization Months |
|---|---|---|---|
| Control Runs | CFSR | CFSR | Apr 2011 – Dec 2017 |
| CS2_IC Runs | CFSR | Same as in control, except initializing the Arctic with CryoSat-2 ice thickness | Oct-Apr 2011-2017 |

## 2.3 Initial Conditions

The control experiments utilized the CFSR reanalysis to initialize both the ice and ocean states. The sea ice concentration and sea ice thickness were obtained at 0.2° resolution, and sea surface temperature and sea surface salinity were obtained at 0.5°
resolution. All these data were interpolated onto the compound model grid.

CryoSat-2 satellite observations have provided another source of estimates for Arctic ice thickness since 2011 during the boreal winter months from October to April (Laxon et al., 2013; Ricker et al., 2014; Grosfeld et al., 2016). In order to explore different initial conditions, additional experiments were conducted where the ice thickness was initialized with CryoSat-2 data, despite the inherent uncertainty in the dataset regarding thin ice (Ricker et al., 2017), as no other datasets were available at
95 that time. The remaining initial conditions, such as snow depth over the sea ice and atmospheric boundary conditions, kept consistent with those used in the control experiments. These experiments utilizing the CryoSat-2 initialization are referred to as "CS2_IC". Since the CryoSat-2 data is confined to the Arctic, the initial sea ice thickness for the Antarctic region in the CS2_IC experiments still relied on the CFSR dataset. Moreover, to maintain consistency in the initialization of both sea ice concentration and thickness, grid points with no sea ice concentration in CFSR or sea ice thickness in CryoSat-2 were assigned
zero values for sea ice concentration and thickness, respectively.

We utilize the monthly CryoSat-2 dataset, where the mean ice thickness data is interpolated onto the CICE model grid as a single thickness category to initialize CICE. We employ an interpolation method that is close to bilinear interpolation, where the ice thickness at each model grid point is calculated as an average of all the raw data points within a radius of 2 grid spacings.

It is important to note that the original CryoSat-2 dataset does not provide ice thickness data in the immediate vicinity of the
105 North Pole. To address this limitation, we estimated ice thickness values at grid points within this region from the surrounding ice thickness data using bilinear interpolation with a bigger search radius (7 grid spacings at 87°N and 10 grid spacings at 89°N). This interpolation process allowed us to fill the data gap near the North Pole with a smoothly varying ice thickness field.

Multiple studies on seasonal forecasts have observed a substantial skill dependence on the choice of the start month (e.g.,
Sigmond et al., 2013; Peterson et al., 2015; Martin et al., 2023). In this study we initialized the control experiments monthly, rather than quarterly, from April 2011 to December 2017. Meanwhile, the CS2_IC experiments, initialized with CryoSat-2 Arctic ice thickness, were carried out monthly during the boreal cold season (October to April) from 2011 to 2017, as presented in Table 1. The integration period for all model experiments was 12 months.

## 3    Model Results and Verification

In this study, the forecast skill was evaluated by comparing the simulated monthly mean of sea ice concentration and thickness with observations and reanalysis data for the matching time period. The evaluation was conducted using root-mean-square error (RMSE) and bias.

All comparisons and verifications are conducted on the native grid, except for overall evaluations at the hemispheric or regional scales. The monthly averages of all fields presented are centered on the stated lead time. Lead times not ending in ".5" are rounded up to the nearest integer month for simplicity (i.e., 11.5-month lead time is rounded up to 12-month lead time).

### 3.1    Hemispheric Scales

Sea ice extent (SIE) is defined as the area of the ocean with at least 15% of sea ice cover, which provides a reliable approximation of the sea ice edge. Goessling et al. (2016) introduced a useful metric for SIE known as the 'Integrated Ice Edge Error', which calculates the integral of all mismatched areas between modeled and observed SIE. This metric offers the advantage of distinguishing between absolute extent error (AEE) and misplacement error (ME). In our study, since AEE predominates over ME, as shown later in Fig. 4, we utilize RMSE and bias to assess SIE. This approach enables a direct comparison with the widely-used Arctic and Antarctic SIE data from the National Snow and Ice Data Center (NSIDC), derived from passive microwave satellite sensors (Meier et al., 2012).

Fig. 1 presents the simulated SIE and sea ice volume (SIV) in the Arctic and Antarctic in the control experiments at lead times of 0.5-month (1st-month average) and 5.5-month (6th-month average) for each target month. It also shows the monthly SIE from NSIDC, SIV observations from CryoSat-2 and SIV reanalysis from the Pan-Arctic Ice Ocean Modeling and Assimilation System (PIOMAS, Zhang and Rothrock, 2003). The curves represent the 2011-2017 average, accompanied by their standard deviations over this time span.

Fig. 1(a) shows the Arctic SIE forecast shows a closer match to NSIDC observations during the warm season compared to the cold season, at both 0.5-month and 5.5-month lead times. The modeled interannual variabilities of the Arctic SIE, indicated by the standard deviation, appear to closely align with those in NSIDC across all seasons. In Fig. 1(b), the Arctic SIV forecast at a 0.5-month lead time has a positive bias relative to the CryoSat-2 observations or the PIOMAS reanalysis during the warm season, and this bias is even more pronounced during the cold season. Moreover, the most substantial positive bias in SIV at a 5.5-month lead time has shifted to the warm season, suggesting that the bias in the SIT initialization during the cold season has persisted for more than one season. The increasing model bias in SIE and SIV with lead time beyond seasonal time scale clearly limits the CICE model's seasonal application.

Fig. 1(c) shows that the modeled Antarctic SIE has a positive bias relative to NSIDC observations across all seasons at a 0.5-month lead time, with the biggest bias occurring during austral spring, similar to the Arctic. A larger positive SIE bias is seen during austral spring at a 5.5-month lead time, resulting in an annual range of SIE that is excessively large compared to observations, approximately a factor of 5 in the Antarctic and a factor of 3 in the Arctic. In Fig. 1(d), the modeled Antarctic SIV exhibits a minimum/maximum in February/September at a 0.5-month lead time, whereas it occurs in March/October at a

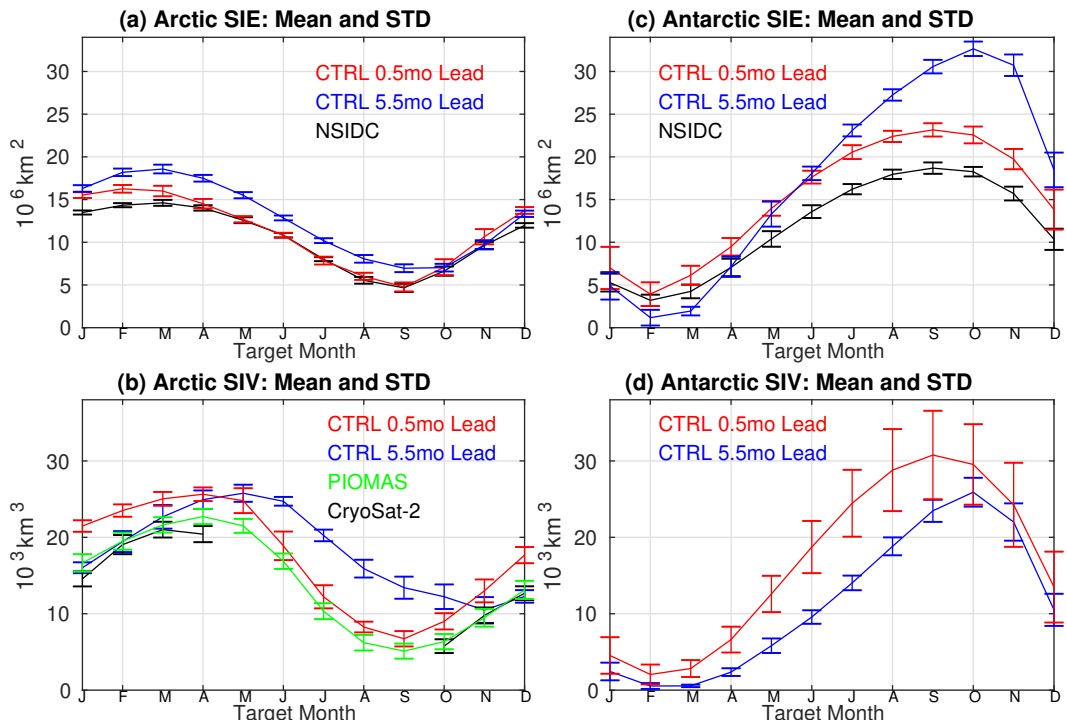

**Figure 1.** Upper: mean and standard deviation of sea ice extent (SIE) at each target month from forecasts at 0.5-month and 5.5-month lead times in the control experiments and NSIDC observations. Averaged over 2011-2017. Lower: same as upper, except for sea ice volume (SIV) with CryoSat-2 observations and PIOMAS reanalysis. Left: Arctic; Right: Antarctic.

5.5-month lead time. The delayed minimum/maximum SIV peak at longer lead times, as is seen in the Arctic, suggests a bias in the modeling system, which remains unclear without Antarctic SIV observation or reanalysis.

The upper panel in Fig. 2 displays the RMSE in Arctic SIE relative to NSIDC observations for the control and CS2_IC experiments, averaged over the period 2011-2017 for each target month at lead times up to 12 months, as well as the difference between the two model experiments. White spaces indicate the absence of model data due to the unavailability of CryoSat-2 data for model initialization between May and September. The results indicate that, in the control experiments, the lowest RMSE in Arctic SIE occurs during summer and fall at all lead times up to 12 months, while the largest RMSE occurs during late winter and early spring at lead times longer than 3 months. In the CS2_IC experiments, where the model is initialized with the CryoSat-2 ice thickness dataset, the RMSE in Arctic SIE is reduced to varying degrees across most seasons. This error pattern in SIE, characterized by pronounced seasonality and insensitive to lead times, is consistent with the outcomes in Wang et al. (2013).

Considering the potential for the RMSE in SIE to be larger during winter due to the extended ice edge and greater ice extent compared to other seasons, it is informative to compare the RMSE with the interannual variabilities of SIE. The interannual variabilities of SIE, measured by the standard deviation, and shown in Fig. 1(a), do not show a clear seasonal dependency at

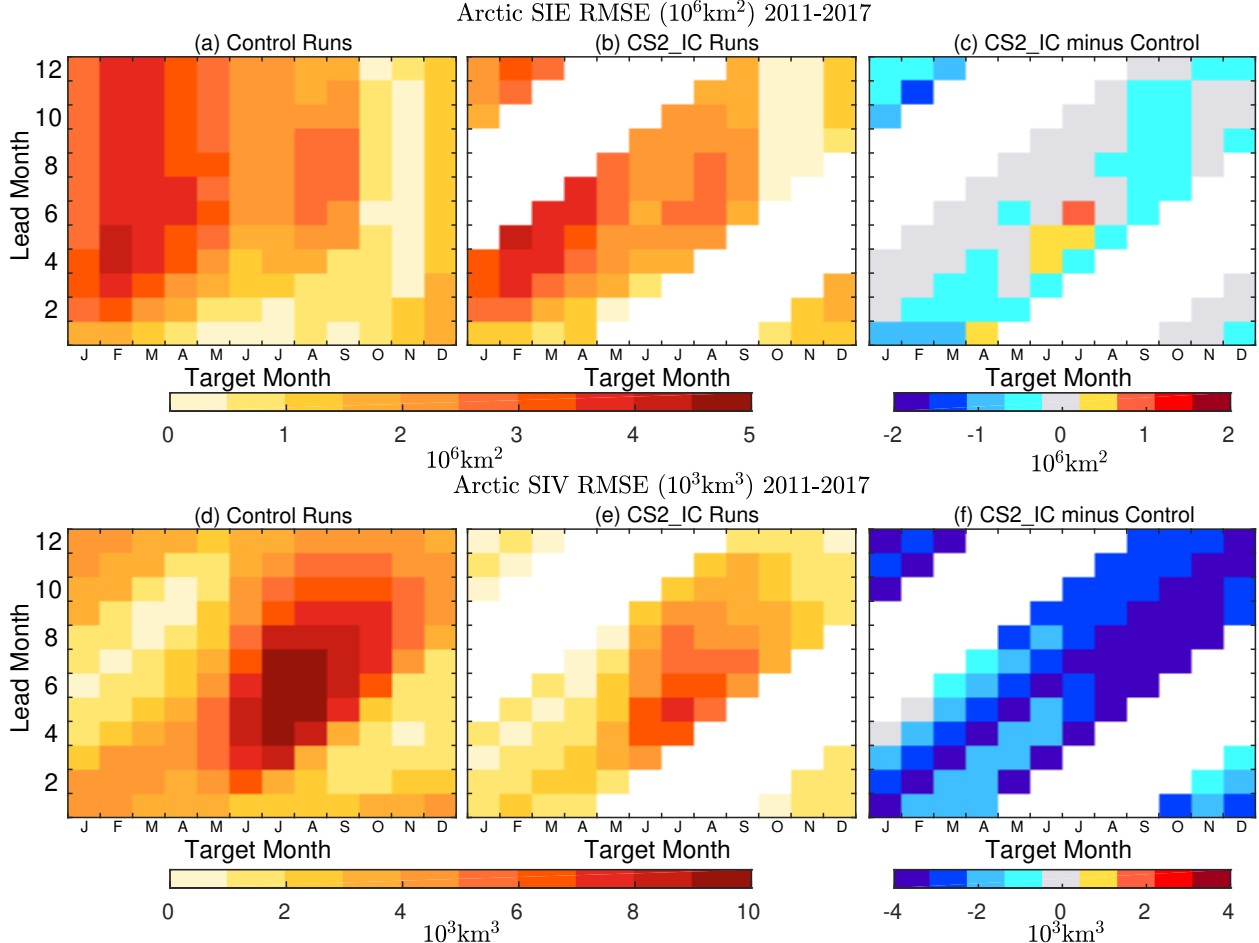

**Figure 2.** Upper: root mean square error (RMSE) in the Arctic sea ice extent ($10^6$ km$^2$) with respect to NSIDC at lead times up to 12 months against the target month in the control [left, (a)] and in the CS2_IC experiments [middle, (b)] and the difference of (b) and (a) [right (c)]. Averaged over 2011-2017. Lower: same as upper, except for sea ice volume ($10^3$ km$^3$) w.r.t. PIOMAS. White spaces indicate no model data due to lack of CryoSat-2 data for model initialization from May to September.

a 6-month lead time, consistent with the NSIDC observations. Nevertheless, the RMSE in the Arctic SIE, shown in Fig. 2(a), reveals a distinct seasonal cycle. The RMSE and the standard deviation for Arctic SIE at a 6-month lead time are comparable in late fall, indicating a higher forecast skill during this season. In contrast, the RMSE substantially exceeds the interannual variabilities in late winter. The decline in forecast skills for late winter is consistent with findings in Peterson et al. (2015); Martin et al. (2023).

The lower panel of Fig. 2 is similar to the upper panel, except it compares SIV to the PIOMAS reanalysis. The results indicate that in the control experiments, SIV is consistently overestimated in all months compared to PIOMAS, with positive bias persisted across most lead times and target months, albeit at varying magnitudes. The most notable bias appears in the

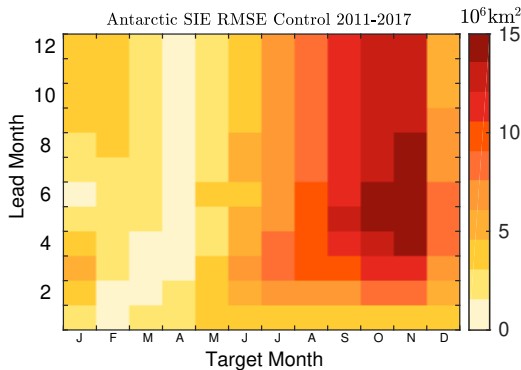

**Figure 3.** RMSE in the Antarctic sea ice extent ($10^6$ km$^2$) w.r.t. NSIDC in the control experiments at lead times up to 12 months against the target month. Averaged over 2011-2017.

'summer barrier' of Fig. 2(d), where skill reemerges prior to winter initialization at a longer lead time. The bias reaches its
minimum magnitude with summer initialization. The CS2_IC experiments show a marked reduction in the overall positive SIV bias, especially in summer and fall with winter and spring initializations.

Despite a broad reduction in SIE and SIV prediction biases through CryoSat-2 ice thickness initialization, the bias pattern in the CS2_IC experiments remains similar to that of the control experiments. The most significant SIV bias occurs in the warm season, while the most substantial SIE bias occurs in the cold season, at lead times of 4-7 months. This pattern is likely related
to the uncoupled experimental setup, where oceanic heat transport is absent. Furthermore, the CryoSat-2 dataset itself may also contribute to the biases, as relative uncertainties are high over thin ice regimes, given that sea ice thickness is determined by the ice surface above the sea level (Ricker et al., 2017).

Fig. 3 presents the RMSE in Antarctic SIE relative to the NSIDC observations for the control experiments during the period 2011-2017. The results indicate that SIE forecasts are in good agreement with observations at all lead times for austral summer
and especially fall when SIE reaches its annual minimum. The highest RMSE in seasonal forecast occurred during austral spring when SIE is at its annual maximum, which aligns with the pattern seen in the Arctic. Additionally, the Antarctic exhibits an 'austral spring barrier' in skill with austral fall and winter initialization, resembling the 'summer barrier' in the Arctic.

Overall, the most pronounced positive bias in SIE occurs during winter in both the Arctic and Antarctic regions. This issue is closely linked to the simplistic representation of ocean dynamics within the utilized mixed-layer ocean model in the standalone
configuration. Here, a one-dimensional column model is employed, without accounting for horizontal ocean transport. This bias is most evident in the Labrador Sea and Bering Sea near the Arctic, as well as in the Southern Ocean surrounding the Antarctic. These regions are particularly influenced by the North Atlantic Deep Water and the Antarctic Circumpolar Current, both integral components of the global thermohaline circulation. Consequently, neglecting oceanic heat transport is likely to lead to a positive bias in SIE, particularly over extended lead times.

## Sea Ice Concentration (%)

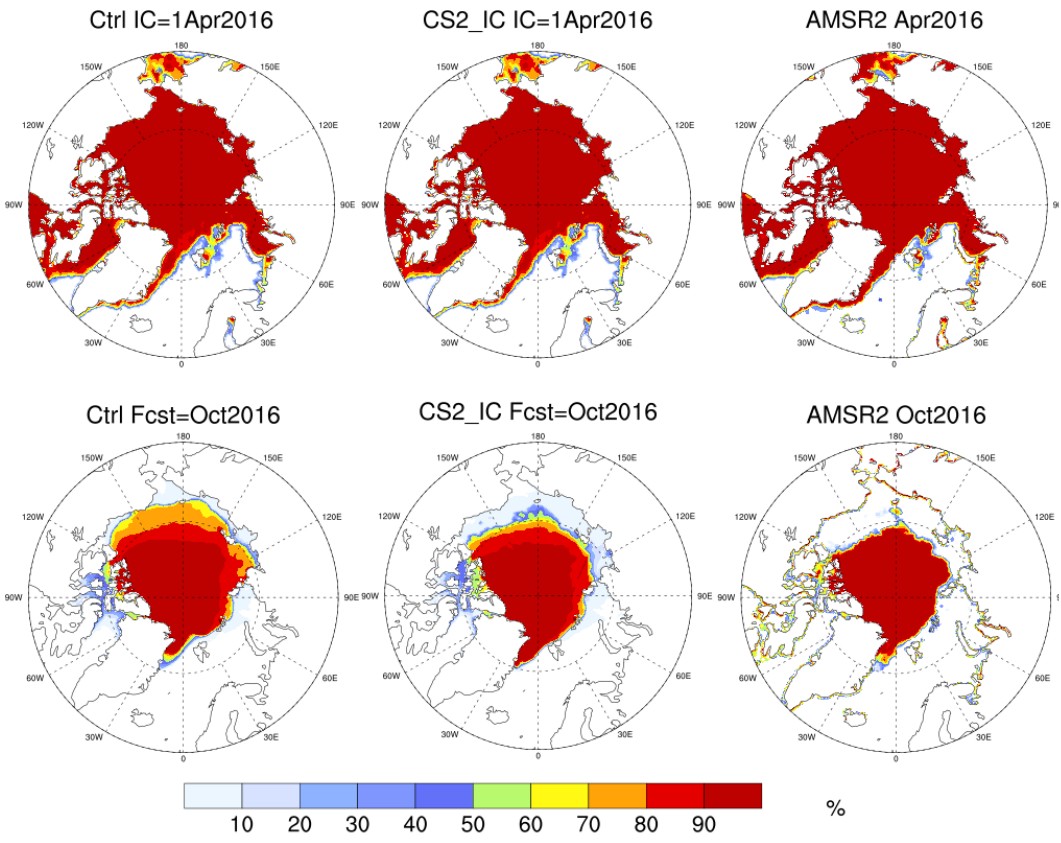

**Figure 4.** Arctic sea ice concentration (%). Left: initial condition on April 1, 2016 and forecast for October 2016 in the control experiment; Middle: same as left, except in the CS2_IC experiments; Right: corresponding AMSR2 observations.

### 3.2 Pan-Arctic forecasts at 6-month lead time

Fig. 4 displays the Arctic sea ice concentration (SIC) in the control and CS2_IC experiments, at initialization on April 1, 2016, the forecast for October 2016, and the corresponding AMSR2 satellite observations (Spreen et al., 2008). The initial SIC in both experiments closely matched the AMSR2 observations. By month 6, the control experiment exhibits a positive SIC bias, primarily in the Beaufort, Chukchi and East Siberian Seas, whereas the SIC from the CS2_IC experiment was closer to AMSR2. This discrepancy can be attributed to the difference in sea ice thickness (SIT) between the two experiments, shown in Fig. 5. Specifically, the control experiment had ice up to 3 m thicker than the CryoSat-2 observations in the region poleward of the Beaufort and Chukchi Seas in the springtime, leading to a positive SIT bias that persisted in this region 6 months later (lower left panel of Fig. 5).

## Sea Ice Thickness (m)

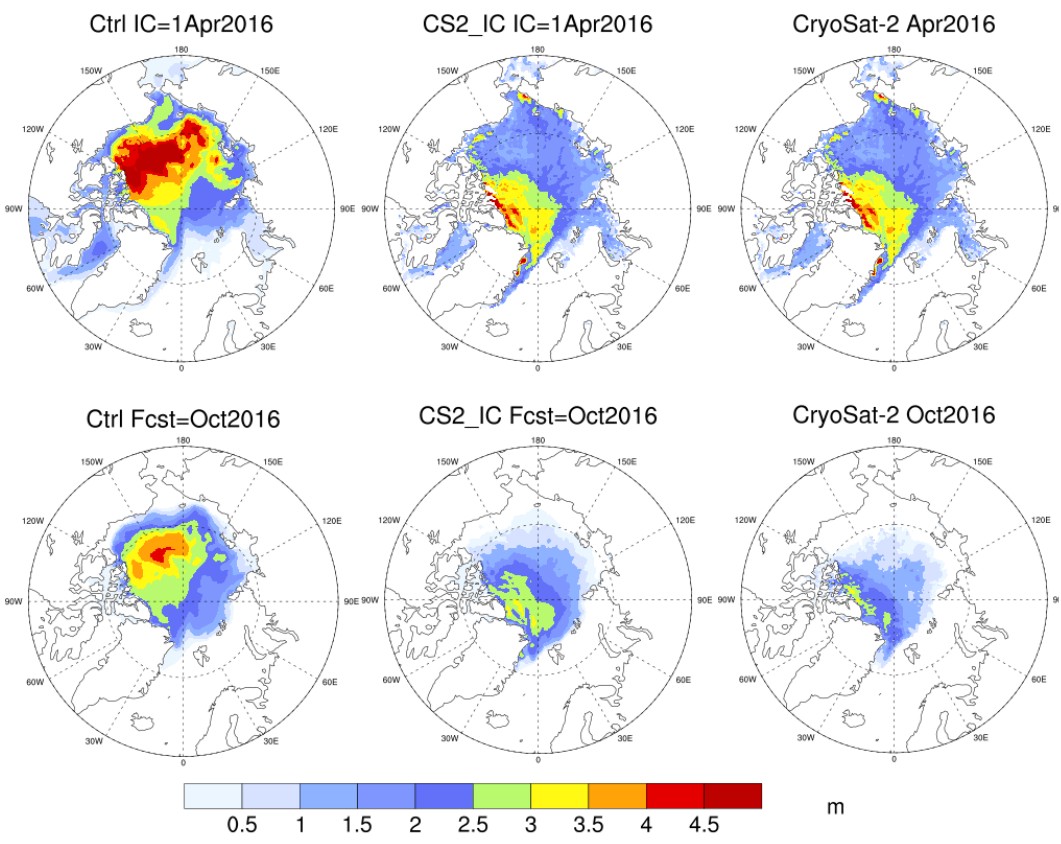

**Figure 5.** Same as Fig. 4, except for sea ice thickness (m) and CryoSat-2 observations.

The positive bias in SIT in the CFSR products presented here is consistent with the findings by Collow et al. (2015). As noted in Saha et al. (2010), SIT is not assimilated in the data assimilation system used in CFSR due to the lack of SIT observations. This bias in SIT can lead to errors in both SIT and SIC in the seasonal forecast, as seen in Figs 4 and 5. The CS2_IC experiment was able to reduce the initial bias in SIT, resulting in improved forecasts of both SIT and SIC. This outcome is in line with previous studies that have shown the positive impact of realistic sea ice thickness initialization on forecast skill (e.g., Day et al., 2014a; Collow et al., 2015; Dirkson et al., 2017; Allard et al., 2018; Blockley and Peterson, 2018; Schröder et al., 2019).

### 3.3 Pan-Arctic 12-month forecasts

Fig. 6 shows the SIT in the initial conditions on January 1, 2016, along with its forecast for December 2016 in both the control and CS2_IC experiments, and corresponding CryoSat-2 observations. The results are consistent with those in Fig. 5, indicating that in the control experiment, the positive bias in SIT in the initial conditions is primarily accountable for the excessive ice

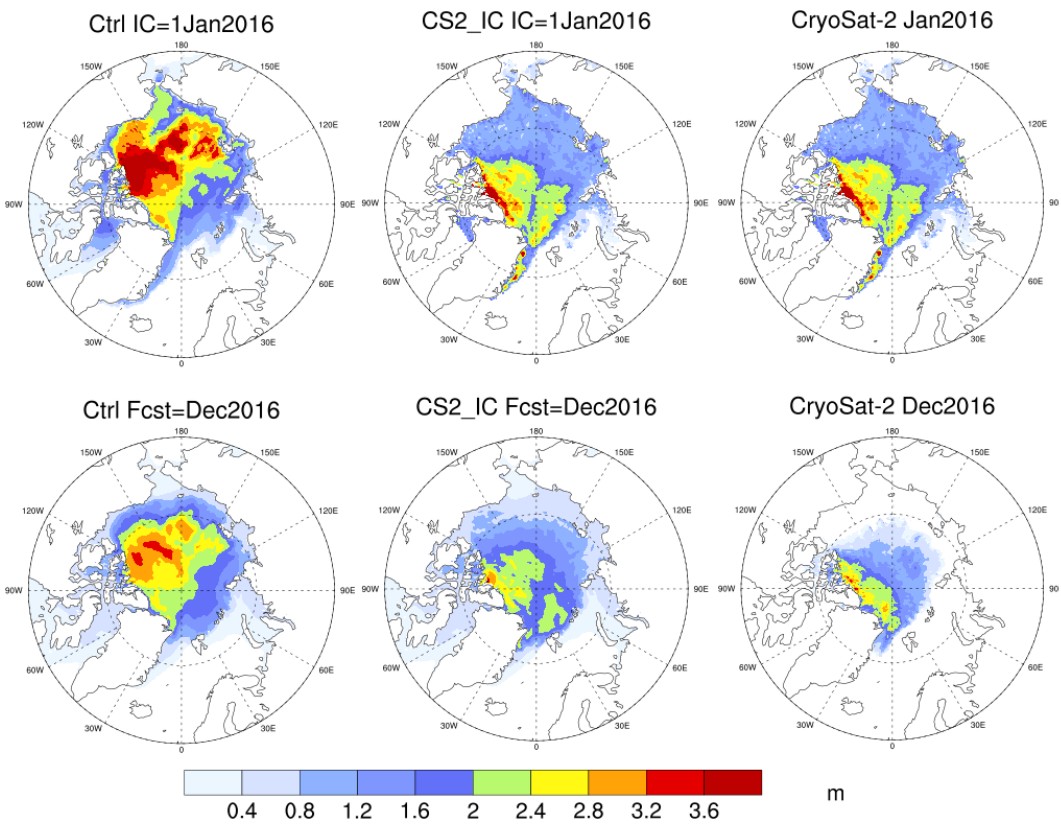

**Figure 6.** Arctic sea ice thickness (m). Left: initial condition on January 1, 2016 (upper) and forecast for December 2016 (lower) in the control experiment; Middle: same as left, except in the CS2_IC experiments; Right: corresponding CryoSat-2 observations.

thickness seen in the Beaufort, Chukchi, and East Siberian Seas in the 12-month forecast. In contrast, the SIT 12-month forecast in the CS2_IC experiment is considerably closer to the CryoSat-2 observations than in the control.

To investigate the mechanism behind the differences between the two model experiments seen in Fig. 6, we present the simulated Arctic SIE and SIV throughout the 12-month integrations in Fig. 7(a) and (b), as well as their comparison to SIE observations from NSIDC and SIV reanalysis from PIOMAS. The initial Arctic SIE in January is similar in both experiments and higher than the NSIDC observations. The initial Arctic SIV in the CS2_IC experiment is lower than that in the control experiments and closer to PIOMAS reanalysis. The SIE and SIV forecasts in the fall in the CS2_IC experiment end up closer to NSIDC and PIOMAS than in the control experiment. The transitions of SIE and SIV from spring to fall show distinctive patterns in these two experiments. The origins of SIE changes can be categorized into two tendencies, originating from thermodynamics

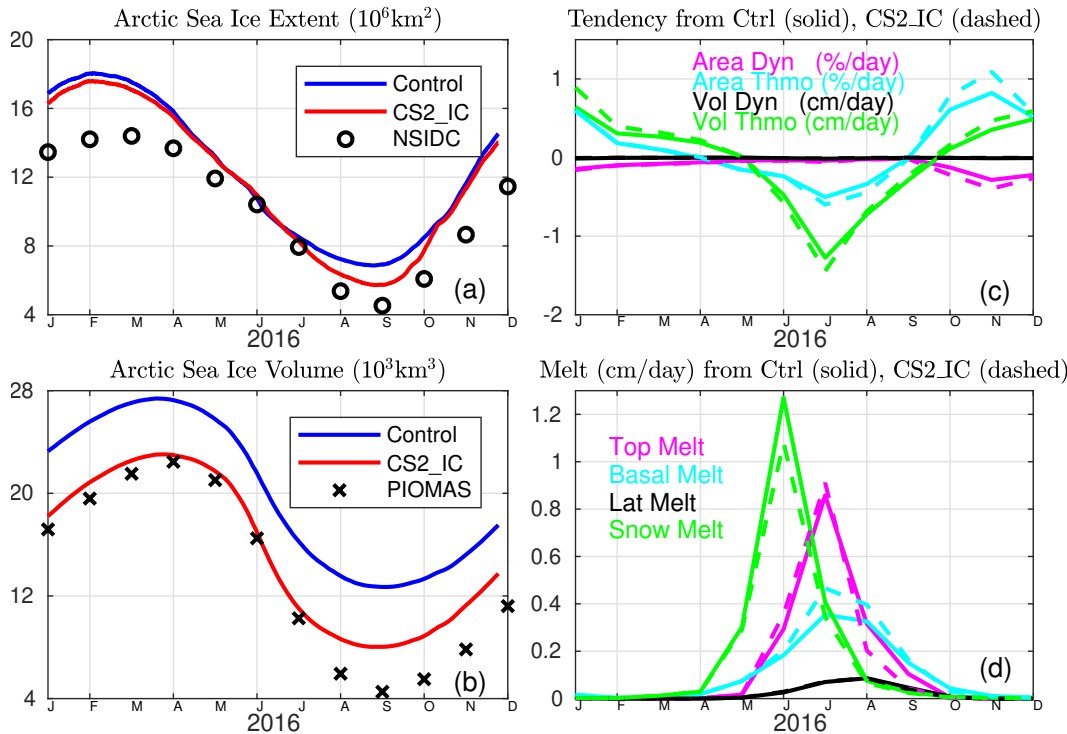

**Figure 7.** Forecasts of Arctic sea ice extent and sea ice volume up to 12 months initialized on January 1, 2016 in the control and CS2_IC experiments are in (a) and (b) in comparison with NSIDC observations and PIOMAS reanalysis. The corresponding tendency of SIE (%/day) and SIV (cm/day) attributed to ice thermodynamics and dynamics are in (c), and melting rates (cm/day) from snow, top, basal and lateral are in (d) in the control (solid) and CS2_IC (dotted) experiments. Averages over area north of 67.5°N are shown in (c) and (d).

and dynamics, respectively. The same applies to SIV tendencies[2]. The ultimate SIE and SIV values result from the interplay of area and volume tendencies arising from both thermodynamic and dynamic processes in the model. These four tendencies

averaged over area north of 67.5°N are shown in Fig. 7(c), where there are more thermodynamics-linked tendencies in both ice area and ice volume than dynamics-linked ones. The negative tendencies in the CS2_IC experiment are slightly larger than those in the control experiment during the warm season. Furthermore, Fig. 7(d) shows the averaged melting rates from snow, top, basal and lateral in both experiments, also over area north of 67.5°N. The snow melting rate peaks one month earlier than the other rates. In July, the top melting rate is about twice as large as that from the basal melt. The most significant difference

in these melting rates between the two experiments is in the basal melt. The CS2_IC experiment, with a higher basal melting rate during summer, yielded both SIE and SIV closer to observations in fall, compared to the control experiment.

To further investigate the difference in melting rates between the two model experiments, we present the simulated ice concentration and thickness in the control experiment in the left column of Fig. 8. The middle and right columns show the

---

[2]The CICE archives use the variable names 'daidtt' and 'daidtd' for area tendencies from thermodynamics and dynamics, and 'dvidtt' and 'dvidtd' for volume tendencies from thermodynamics and dynamics, respectively.

differences in sea ice concentration, sea ice thickness, top and basal melting rates between the CS2_IC and control experiments. These variables are forecast for July 2016 from the runs initialized on January 1, 2016. One dominant feature in the Arctic between 120°E and 120°W is that both ice concentration and thickness are smaller in the CS2_IC experiment than those in the control experiment. This is consistent with a higher melt rate, both at top and bottom, in the marginal ice zone in the CS2_IC experiment, which appears to be more realistic than in the control experiment.

There is still a positive bias in SIT in the Greenland Sea and central Arctic in the CS2_IC experiment as shown in Fig. 6. Fig. 9 compares the modeled monthly mean SST with the OISSTv2 dataset (Reynolds et al., 2007) for target months April, August and December, at lead times of 4, 8 and 12 months, in the CS2_IC experiment initialized on January 1, 2016. During the cold season, the modeled SST shows a cold bias dominating the Greenland, Iceland, and Norwegian Seas (GINS) as well as the Bering Sea. During the warm season, although there is a warm bias away from the ice edge, a cold bias still dominates the marginal ice zone. This cold bias in the marginal ice zone is attributed to the lack of northward heat transport to the Arctic from the Atlantic and Pacific Oceans, as discussed in Sec. 3.1. Meanwhile, the positive bias in SIT in the central Arctic could be linked to the colder atmospheric forcings from the CFSR dataset, which are utilized to drive the CICE model in this study.

In summary, the CICE model's seasonal prediction of the Arctic SIE has higher skill in the warm season than in the cold season at almost all lead times in the control experiment. The largest Arctic SIT bias occurs in summer at 3- to 6-month lead times and in fall/winter at roughly 6- to 9-month lead times, mostly due to a thicker ice initialization in the cold season and insufficient melting rates in the warm season in the control experiments. Using a more realistic initialization of thinner ice in the Arctic, incorporating CryoSat-2 observations in the CS2_IC experiments, enhances forecast skill in both SIE and SIV at all lead times. However, a positive SIV bias still exists in the CS2_IC experiments during summer at 3- to 6-month lead times. Further examination of specific regions in the following section will provide additional insights into this.

## 3.4 Regional Scales

Analysis of the sea ice prediction skill was conducted in regions defined in Fig. 10. For the purpose of this study, we refer to the combination of the Barents, Kara, and Greenland Seas as the BKG seas. To investigate the seasonality of the forecast skill seen earlier, regional SIE and SIV forecasts in the control experiment (solid blue) and the CS2_IC experiment (dashed red), as well as AMSR2 observations (black circle) were presented in Fig. 11. These forecasts covered 12-month integrations initialized on April 1, average of 2013 to 2017, the period when AMSR2 observations were available. In Fig. 11(a), SIE in the control experiments was reasonably predicted in most regions in summer and fall, but overpredicted in the BKG Seas, Baffin Bay and the East Siberian Sea in winter at lead times longer than 6 months. The CS2_IC experiment, initialized with more realistic SIV, showed improved SIE prediction in the East Siberian Sea at lead times above 6 months. However, there were no notable changes seen in the BKG seas and Baffin Bay.

The corresponding SIV forecasts in these regions from the same experiments were also analyzed, and the results were presented in Fig. 11(b), along with CryoSat-2 observations. The positive SIV bias in the Beaufort, Chukchi, East Siberian and Laptev Seas seen in the control experiment was largely reduced in the CS2_IC experiments at almost all lead times, indicating that the SIT bias in initialization was the primary cause of the positive SIV bias in the seasonal forecast in this Arctic region

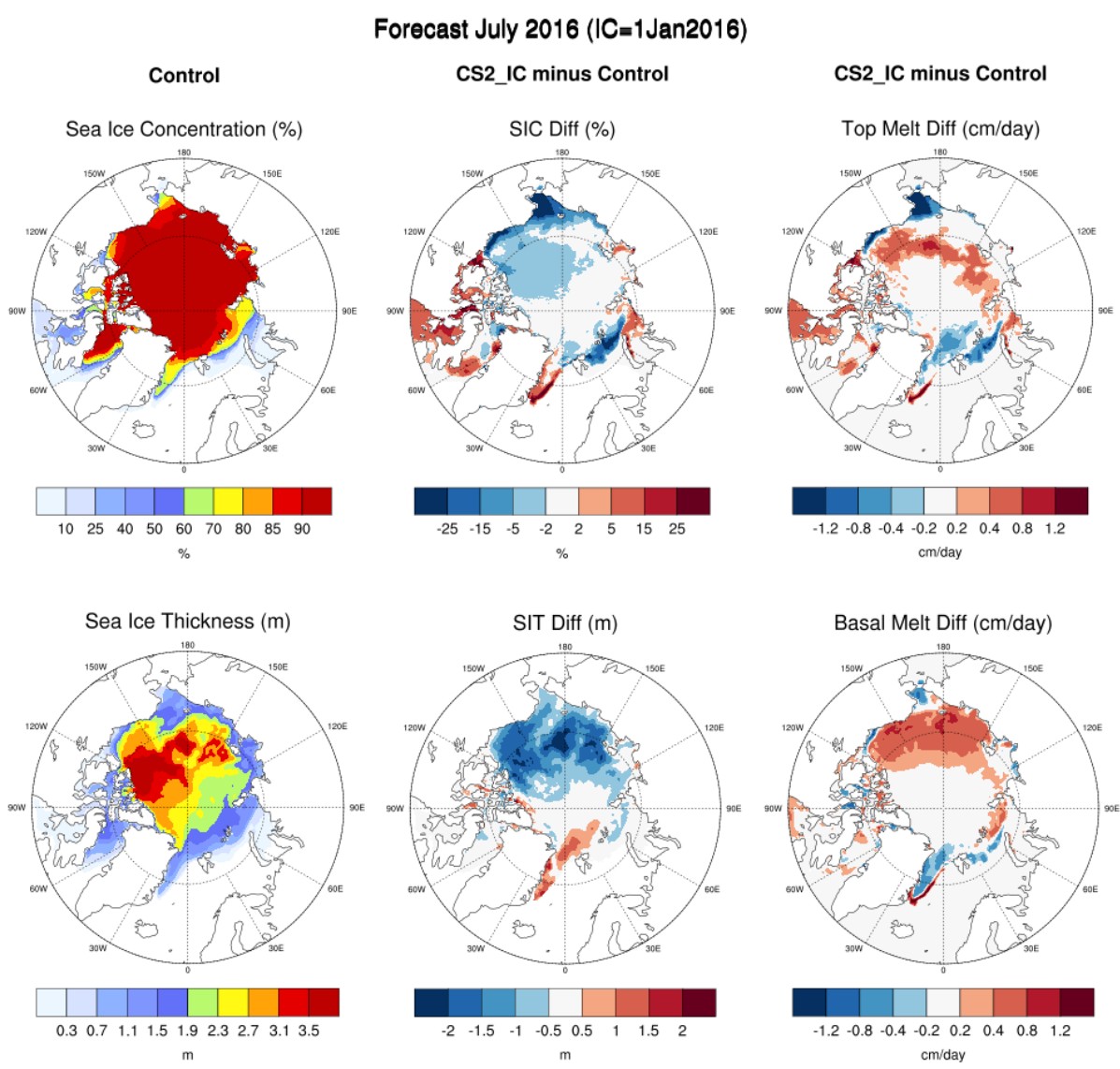

**Figure 8.** Arctic sea ice concentration (%) and sea ice thickness (m) in the control experiment are shown in the left column. The difference between CS2_IC and control experiments in the sea ice concentration (%), sea ice thickness (m), top and basal melting rates (cm/day) are in the middle and right columns. All are forecast for July 2016 initialized on January 1, 2016.

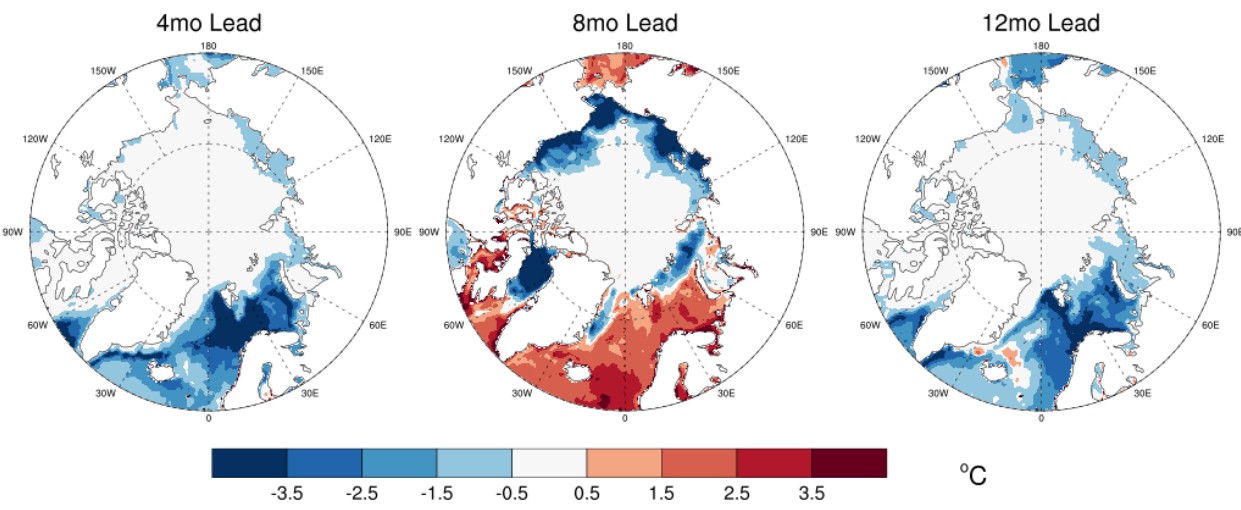

**Figure 9.** SST bias against OISSTv2 at lead times of 4, 8 and 12 months, respectively, in the CS2_IC experiment initialized on January 1, 2016.

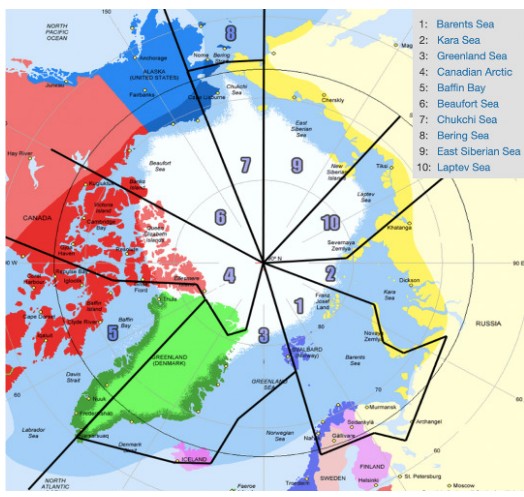

**Figure 10.** Different regions in the Arctic Ocean. Reproduced from https://arctic-roos.org.

between 120°E and 120°W. However, positive biases in SIE in the BKG Seas and Baffin Bay, located in the vicinity of the Atlantic Ocean, remain unchanged.

Fig. 12 presents the regional bias in SIE and SIV, similar to Fig. 11, with model initializations in October, when Arctic sea ice coverage is near its minimum. During winter and spring, when observations are available, the SIE and SIV predictions closely align with AMSR2 and CryoSat-2 observations in most regions. However, both the control and CS2_IC experiments exhibit a positive bias in SIE and SIV in the BKG Seas, Baffin Bay, and the East Siberian Sea. This finding is consistent with Fig. 11, which used April initializations.

Comparing the CS2_IC experiments with the control experiments, there is a modest improvement in SIV skill in the Arctic region between 120°E and 120°W. This improvement arises from initializing with a more realistic SIV, as opposed to the control experiments. The benefit of a more realistic ice thickness initialization is more pronounced when initialized from a high SIC state in April, compared to when initialized from a low SIC state in October, since SIV in the CFSR dataset has a bigger bias in April than in October.

The experiments with both April and October initializations revealed a positive bias in SIE and SIV in the BKG Seas and Baffin Bay. This bias was attributed to inadequate simulation of the interaction between these regions and the Atlantic Ocean, primarily caused by the use of a simplified mixed layer model, as mentioned earlier. The poleward ocean heat transport from both the Atlantic and Pacific Oceans has a significant impact on Arctic sea ice, as shown in Docquier and Koenigk (2021). In the model, the absence of northward oceanic heat transport through the Barents Sea Opening, Fram Strait, Davis Strait (Atlantic

gateways), and Bering Strait (Pacific gateway) results in the sea ice edge extending too far south during winter.

## 4 Summary

This study focuses on assessing the trend-independent skill of sea ice prediction at seasonal time scales in the CICE sea ice model in standalone mode. The model is driven by the atmospheric forcings from the NCEP CFSR reanalysis, coupled with the built-in mixed layer ocean model in CICE. The control experiments are initialized with the CFSR reanalysis for both ocean

and ice states with multiple year-long experiments. We aim to identify biases that limit its suitability for seasonal prediction.

The model demonstrates commendable forecasting performance for SIE during the warm season in both the Arctic and Antarctic, with lead times up to 12 months. However, a significant bias emerges in the SIE seasonal forecast during boreal late winter and early spring in the Arctic, as well as austral spring in the Antarctic. These biases limit the seasonal prediction skill of the CICE model.

We identified the first-order bias as the positive SIT in the initialization of the control experiments. The initial SIT from CFSR is consistently higher than the CryoSat-2 satellite observations, and the excessive ice thickness is often retained for more than one season. We were able to reduce this bias in the CS2_IC experiments by initializing Arctic sea ice thickness using the CryoSat-2 satellite observations while keeping everything else unchanged. Although the CryoSat-2 ice thickness data are only available from October to April in the Arctic and have bias when ice thickness is small, the multi-year experiments initialized

with this dataset clearly show the bias reduction in both SIE and SIV at most lead times in almost all seasons.

(a) Ice Extent ($10^6$km$^2$, IC=1 Apr 2013-2017)

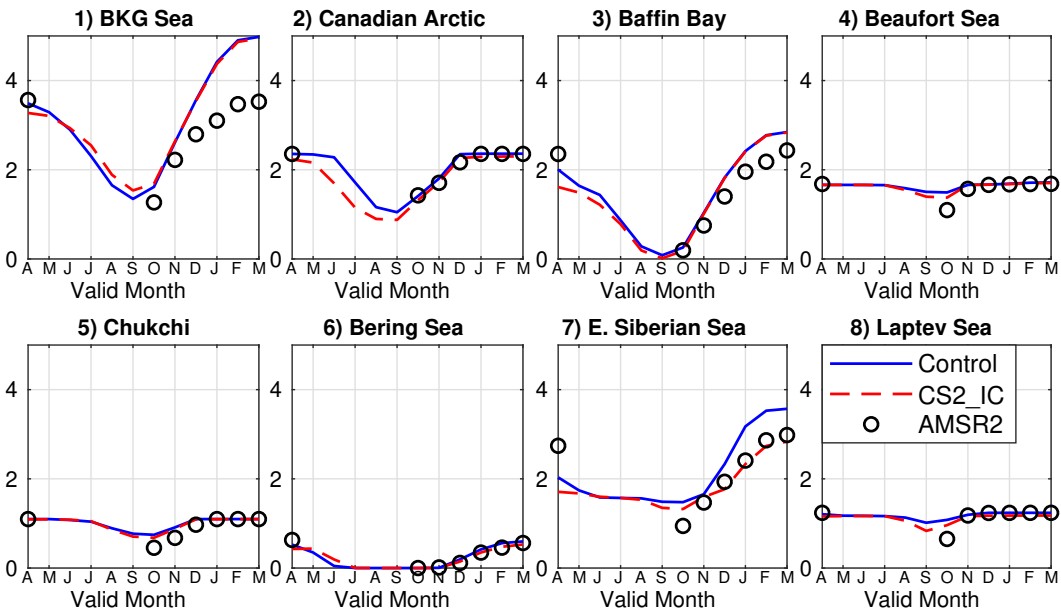

Ice Volume ($10^3$km$^3$) IC=1 Apr 2013-2017

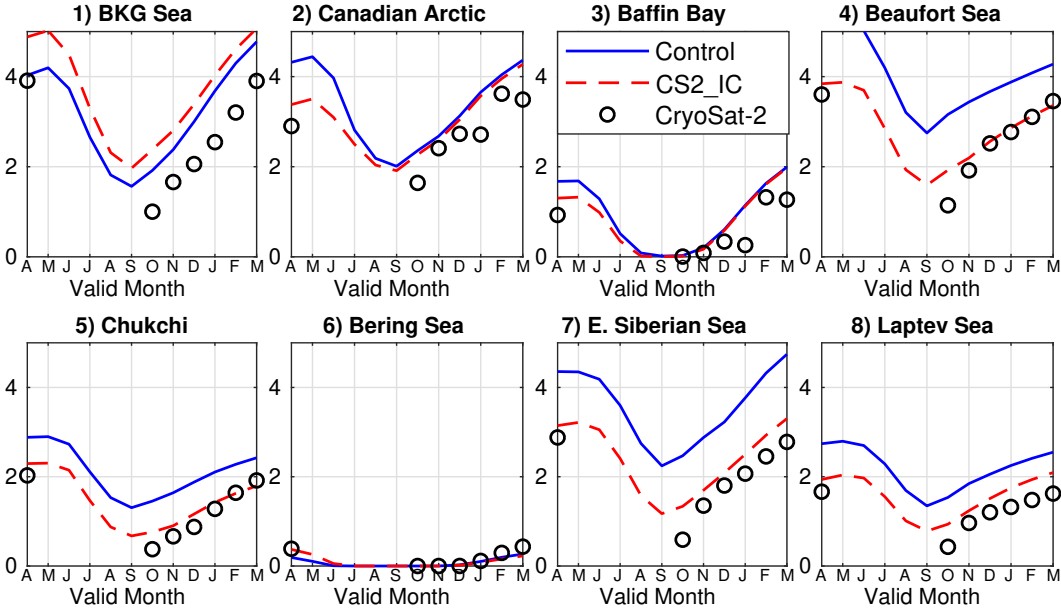

**Figure 11.** Sea ice extent and thickness during 12-month integration in the control (solid blue) and CS2_IC (dashed red) experiments, initialized on April 1, 2013 to 2017, as well as AMSR2 and CryoSat-2 observations (circles) in each region where (1) represents Barents, Kara and Greenland Seas (BKG) combined.

(a) Ice Extent ($10^6$km$^2$, IC=1 Oct 2013-2017)

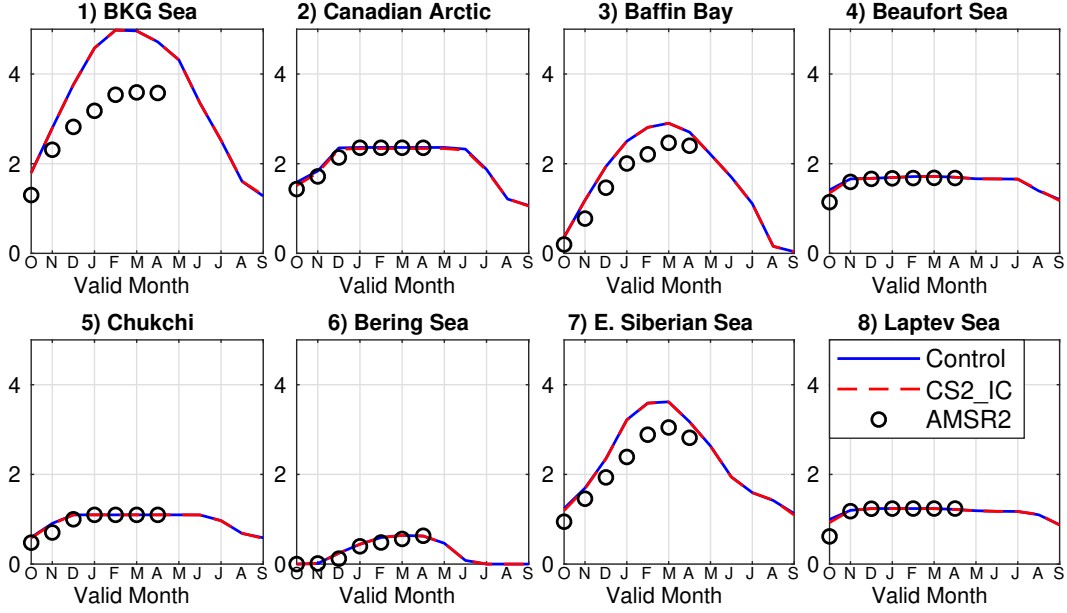

(b) Ice Volume ($10^3$km$^3$, IC=1 Oct 2013-2017)

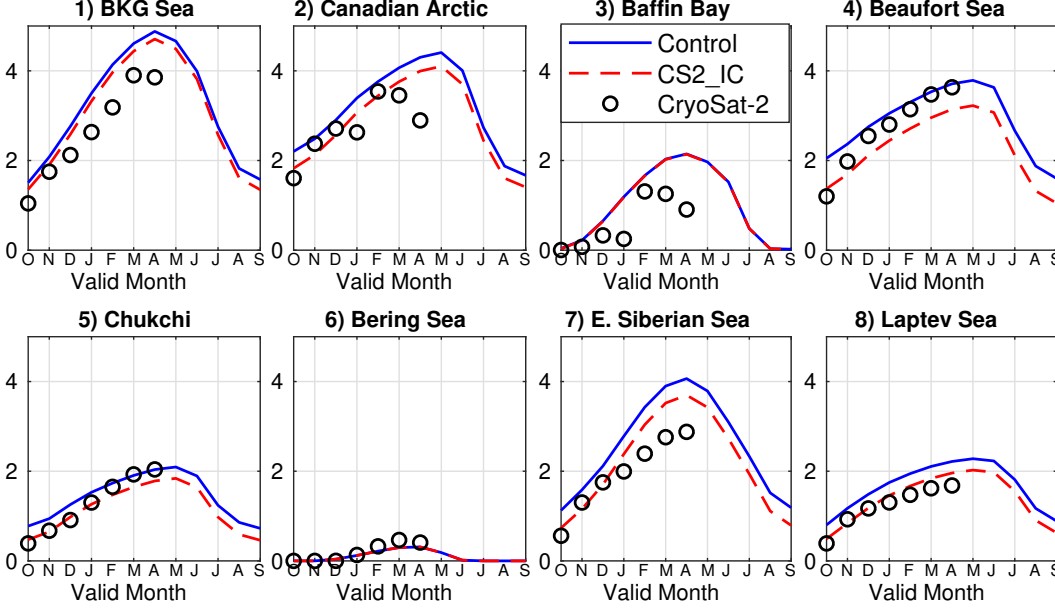

**Figure 12.** Same as Fig. 11, except with experiments initialized on October 1, 2013 to 2017.

Following the mitigation of the positive SIT bias, our analysis reveals a cold bias in SST in the CS2_IC experiments near the marginal ice zone. This bias is attributed to the simplified column mixed-layer ocean model employed in the experiments. The column model neglects the northward heat transport from both the Atlantic and Pacific Oceans, resulting in a cooler SST primarily around the periphery of the Arctic Ocean. A similar scenario arises in the Antarctic. Despite this limitation, the model is capable of delivering a decent SIE forecast, particularly during the warm season at all lead times up to 12 months.

Another source of bias may be from the CFSR atmospheric forcings used to drive CICE in this study, potentially leading to a positive SIT bias seen in the central Arctic. Other possible sources for biases could arise from uncertainties in the CryoSat-2 dataset over thin ice, or limitations in the CICE model itself. To address both issues, the next step is to incorporate CICE in a fully coupled atmosphere, ocean and sea ice system, initialized with the sea ice thickness from Ricker et al. (2017).

## 5 Conclusion

A standalone sea ice model test serves as an essential tool for understanding the physical properties of sea ice and its response to changing environmental conditions. The results presented here are consistent with those in the current NOAA operational seasonal forecast model CFSv2 (Wang et al., 2013), which serves as the coupled model responsible for producing the atmospheric reanalysis used to drive the CICE experiments in our study. However, the RMSE shown in Fig. 2 appears larger than that reported in Wang et al. (2013) for several reasons. Firstly, the latter is based on a fully coupled model allowing feedback between atmosphere, ocean and sea ice, whereas our study uses an uncoupled sea ice model with prescribed boundary conditions that may not always align with the sea ice model. Secondly, both the ocean model and the sea ice model used in the two studies differ, particularly with the limitation from the one-dimensional mixed layer ocean model used here. Finally, the validation period also differs, with Wang et al. (2013) comparing to the 26-year climatology of 1981-2007, while the 2011-2017 period used in our study is characterized by a smaller observed sea ice area than in earlier decades.

This study highlights the importance of accurate ice thickness initialization for seasonal sea ice prediction. Given the limited availability of sea ice thickness observations in both time and space, none of the current operational seasonal prediction systems incorporate sea ice thickness observations into their data assimilation system. Our study, which demonstrated enhanced skill in SIE following the transition to a more realistic sea ice thickness initialization, underscores the untapped potential skills of such practices. These findings in the seasonal prediction simulations are consistent with prior research in the climate community. Therefore, data assimilation of sea ice, including sea ice coverage and thickness, either from observations or reanalysis, appears highly relevant for advancing seasonal prediction skill. Additionally, this study emphasizes that the suitability of CICE for seasonal prediction relies on various factors, including initial conditions such as sea ice coverage and thickness, as well as atmospheric and oceanic conditions like oceanic currents and SST.

*Author contributions.* SS and AS designed the study, performed the analyses and wrote the manuscript. SS carried out the numerical experiments and did the first draft of the analyses.

*Competing interests.* The authors declare that they have no conflicts of interest.

*Acknowledgements.* This research is funded by the Global Model Test Bed at Developmental Testbed Center under NGGPS. We thank David Bailey for helpful discussions and sharing a script to convert initial conditions to restart conditions for CICE, Xingren Wu for suggesting us-

330 ing a mixed layer ocean instead of SST from CFSR and Rainer Bleck for constructive suggestions during the interval review. Discussions with James Rosinski, Ola Persson, Julie Schramm, Janet Intrieri, Chris Fairall, Antonietta Capotondi and Ligia Bernardet are also much appreci- ated. Benjamin W. Green helped with graphics. The authors would like to express their gratitude to the three anonymous reviewers for their valuable comments, which improved the quality of this manuscript. CFSR data used here are from the Research Data Archive, managed by the Computational and Information Systems Laboratory at the National Center for Atmospheric Research in Boulder, Colorado. The ice concen-

335 tration data is obtained from the National Snow and Ice Data Center (NSIDC; http://nsidc.org/data/docs/daac/nsidc0051_gsfc_seaice.gd.html).

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
