# Peer review of "Suitability of CICE Sea Ice Model for Seasonal Prediction and Positive Impact of CryoSat-2 Ice Thickness Initialization"

_The Cryosphere, 2023_

## Referee Comment (RC2)

**Review of: Suitability of CICE Sea Ice Model for Seasonal Prediction and Positive Impact of CryoSat-2 Ice Thickness Initialization**

by Shan Sun and Amy Solomon

November 30, 2023

**Manuscript Synopsis**

This manuscript evaluates the usage of the CICE sea ice model for seasonal forecasting in NOAA's Unified Forecast System by examining 12 month simulations of CICE driven by reanalysis atmospheric forcing and an ocean 1-D mixed layer model. Using root mean square errors in integrated quantities, the authors evaluate hemispheric wide errors in sea ice extent and sea ice volume – but not spatial errors.

**My evaluation of this manuscript is to reconsider after major revisions.** These major revisions (major comments 1, 2 and 4 below) can mostly be dealt with by the supply of additional information either in text or in response. Major comment 3 is a recommendation which could enhance the manuscripts applicability, but is not critical to its publishability.

The most pressing issue, which may stem solely from my lack of understanding of running CICE in stand-alone mode and comes down to a lack of details with respect to the oceanic boundary condition (Section 2.2). While I understand the authors usage of the mixed layer ocean model for the sea ice thermodynamics, it does not explain what was used for bottom boundary dynamical conditions – and leads me to believe there are none, or more precisely, it assumes an unmoving ocean. Perhaps this could be resolved with some rather simple re-working of the model introduction (i.e. more to Section 2.2 then just the thermodynamic lower B.C.). As the location of the thickest ice is predominantly set by dynamic processes (driven against the Canadian Archipelago and northern Greenland) – and the location of the thickest ice in the model setup is not modelled very well (in the Beaufort Sea and central Canada Basin), this suggests a possible deficit in dynamical tendencies that requires a better articulation of the dynamical lower boundary conditions. [Scientific Quality: Methods not adequately explained.]

A 2nd major omission of the manuscript is the mistaken impression implied in the introduction (ll. 37-41; and granted it is not stated explicitly) that CICE has mainly been used for climate simulations. This is far from the truth, as CICE is already in use in many operational seasonal, sub-seasonal and NWP systems, which I detail further below. [Originality: Manuscript does not adequately present current scientific understanding.]

Lastly, while the skill metrics used in the manuscript are acceptable – but very climate oriented – they can be enhanced quite easily to assess integrated errors as opposed to errors in integrated quantities (IIEE), simply by changing the order in which the operations are performed. This point I would be willing to give the authors some leeway with. The regional results (Section 3.4) do address this issue somewhat – although Figure 11 is not correct, but Figure 12 does not suggest any cancellation of error (only deficits in sea ice). Nevertheless it could highlight further errors not elicited by Figures 2 and 3. [Scientific Quality: Validation methods could be improved. Significance: Application of results could be improved through enhanced spatial information.]

I appreciate that this manuscript may already have been through at least one round of review and revision, so I hope my suggestions do not pose too onerous a task.

**My recommendation is Reconsider after Major Revision.**

**Major Comments**

1. Section 2.2: No explanation is given for lower (ocean) dynamic boundary condition. While the authors dedicate a sub-section (Section 2.2) to the ocean boundary conditions, this only explains the lower boundary condition for Sea Surface Temperature (SST) – which is chosen to be a mixed layer 1-D ocean – and would only effect the model thermodynamics. No explanation for specification of the dynamical boundary conditions, or more precisely, the ocean surface currents is given. Presumably, this would have a large effect on the sea ice dynamics, which it is impossible to make informed decisions regarding without further information.

2. ll. 37-41: The introductory paragraph gives the false impression that the CICE sea ice model is primarily used for climate simulations (Note: I do not deny that the model was initial constructed for this purpose), implicitly implying that its introduction as the sea ice component for NOAA's Unified Forecasting System is a novel usage. The exact phrasing used (ll 39-40) "its suitability for seasonal forecasting needs to be assessed." I do agree, or at least do not disagree with that statement, however, some credit through citation is deserving to the multitude of operational (and quasi-operational) systems currently in use for seasonal forecasting use throughout the world for over a decade (Note: many of these, by necessity, are self-serving):

   (1) UK Met Office GloSea4/5 system (`https://doi.org/10.1175/2010MWR3615.1`, `https://doi.org/10.1007/s00382-014-2190-9`, `https://doi.org/10.1002/qj.2396`)

   (2) Korea Met Agency version of GloSea5 (`https://doi.org/10.5194/tc-2018-217`)

   (3) ASSESS-S1/2, Australia Bureau of Meteorology version of GloSea5 (`https://www.publish.csiro.au/es/ES17009`, `https://www.publish.csiro.au/ES/ES22026`).

   (4) CanSIPSv2 GEM-NEMO-CICE component (`https://doi.org/10.1175/WAF-D-19-0259.1`, `https://doi.org/10.1175/WAF-D-22-0193.1`, `https://www.tandfonline.com/doi/pdf/10.1080/07055900.2023.2252387`

   6 of 11 dynamical contributors to the ARCUS July 2023 sea ice outlook (`https://www.arcus.org/sipn/sea-ice-outlook/2023/july` (5 listed below as UK Met Office is one of 6)

   (5) RASM/NPS (`https://doi.org/10.5194/gmd-11-4817-2018`)

   (6) ArcIOAM, National Marine Environmental Forecasting Center, China (`https://doi.org/10.5194/gmd-14-1101-2021`)

   (7) FIO-ESMv1.0, Qingdao, China (`https://doi.org/10.3389/fmars.2020.00504`

   (8) FGOALS-f2 V1.3, Institute of Atmospheric Physics, China (`https://doi.org/10.1029/2019MS002012`)

   (9) Unified Forecast System, NOAA (`https://doi.org/10.1029/2022GL102392`). Note: This is the prototype system being evaluated in this manuscript.

   and a final system via a google search for CICE seasonal forecasts

   (10) SLAV/INMIO/CICE, Marchuk Institute of Numerical Mathematics / Shirshov Institute of Oceanology / Hydrometeorological Centre of Russia (`https://doi.org/10.1515/rnam-2018-0028`)

   to which I will also add shorter range S2S, monthly and short range ($< 10$day) systems:

   (11) Global Ensemble Prediction System, Environment and Climate Change Canada (`https://doi.org/10.1002/qj.4340`) (S2S/monthly/extended)

   (12) Global Ice Ocean Prediction System / Regional Ice Ocean Prediction System, Environment and Climate Change Canada (short range) (`https://doi.org/10.1002/qj.2555`, `https://doi.org/10.1175/MWR-D-17-0157.1`, `https://doi.org/10.5194/gmd-14-1445-2021`)

*(13)* Forecasting Ocean Assimilation Model (FOAM), UK Met Office (`https://doi.org/10.5194/gmd-7-2613-2014`) (short range)

*(14)* Prototype UK Met Office Coupled System for NWP (`https://doi.org/10.1175/WAF-D-20-0035.1`) (short range)

While I would not expect (the list was longer than even I had initially assumed!) the authors to cite each and every one of these, it would still be appropriate to underline the usage of the CICE model in existing operational systems – and highlight and reference their results against the sea ice predictability of some of these earlier systems, particularly when an assessment of sea ice performance has been undertaken. See point 4 for one such possible connection.

3. RMSE skill measure: While the RMSE quantification of error for the hemispheric domain is adequate, it is also relatively non-standard. More usual to be found in seasonal papers is anomaly correlation (`https://doi.org/10.1002/grl.50129`, `https://doi.org/10.1007/s00382-014-2190-9`), which eliminates bias. However, more modern skill estimates account for both the area of the sea ice extent along with its position, through skill measures like Integrated Ice Edge Error (`https://doi.org/10.1002/2015GL067232`). Implementation of this would be simple enough. All you need do is commute the order in which the area and rmse error operations are performed. [I.e. calculate the square error of ice existence $(M - O)^2$, where $M$ is modelled ice $> 0.15$ concentration and $O$ is observed ice $> 0.15$ concentration in any grid cell, and then perform your summation. Note: Since $(M - 0)$ is 1/0 it does not matter whether you square or take absolute value. Taking the square easily allows you to generalize for an ensemble, where M is replaced with P, the fraction of ensemble members with ice (`https://doi.org/10.1002/qj.3242`). You could also consider replacing the RMSE of total ice volume with the integrated square error of grid cell ice volume (it is preferable to add square error, not root mean square error). The latter will then give you a double penalty for having ice volume in the Beaufort Sea, but little over the Canadian Archipelago and Greenland. I will not insist on the authors doing this, but it could enhance the applicability of their results.

4. Although it is also a characteristic of the IEEE as well (so that will not solve this problem), the RMSE of an area integrated quantity will be inherently larger when that integrated quantity is larger. Thus it may be natural for the RMSE in February and March to be large solely because the ice extent is large during those periods. It would be a more accurate assessment of whether the ice area predictability is better or worse by comparing the RMSE with the interannual variability for that time of year. That being said, deterioration of predictability seems also to occur for mid to late winter in many other assessments of sea ice predictability through correlation skill assessments (`https://doi.org/10.1002/grl.50129`, `https://doi.org/10.1007/s00382-014-2190-9`, `https://doi.org/10.1175/WAF-D-22-0193.1`). Perhaps the authors can comment on this – and cite previous seasonal sea ice skill assessments.

5. Figure 11: Figure 11 is an error/omission. Figure 11 is identical, save for 6 month offset x-axis to the correctly attributed Figure 12.

**Minor Comments**

1. Section 2.1: You should emphasize that you are forcing the sea ice integration with "0-hour lead" reanalysis forcing. In other words, you are not performing a true seasonal forecast, where the forcing is also of long lead time.

2. Section 2.2: The above point then begs the question as to why "0-hour lead" SST forcing does not lead to a better ice concentration (not necessary thickness) integration as described in Guemas et al (2014; http://dx.doi.org/10.1007/s00382-014-2095-7). Perhaps it did – the explained reason for abandoning due to it "result(ing) in an unrealistic increase in basal melt," was for reasons of unrealistic thermodynamics, it may still have resulted in a more accurate sea ice concentration integration – likely at the cost of a more unrealistic sea ice thickness integration. Perhaps the authors can expand their explanation.

3. Section 2.3: It is **very** important you explicitly specify you initialized to the Cryosat-2/SMOS dataset (Ricker et al, 2014, `https://doi.org/10.5194/tc-11-1607-2017`). Otherwise readers will be confused on how you initialized sea ice with thickness less than 1m.

**Minor Presentation Comments**

1. Perhaps this is pedantic, but the units should really be on the colour bar (Figures 2, 3, 4, 5, 6, 8, 9), or on the y-axis (Figures 7, 11, 12) if possible, and not just (could be additionally) in the figure title. Note: Figure 9, lacks units completely – although fairly obviously °C/K. The latter at least needs to be corrected.

---

## Author Comment (AC1)

We thank the reviewer for their careful reading of the manuscript and constructive remarks, which helped improve the quality of the manuscript. We have addressed all the recommendations, see details below (reviewer's comments in black, our replies in blue).

The Los Alamos sea ice model (CICE) is tested in standalone mode in the study and the performance on seasonal sea ice prediction is examined. The paper is well written and organized with clear logics that readers can readily follow. The results present now is informative. However, one more aspect can be considered to further improve the manuscript. My comments are generally listed below:

I would recommend add "Summary" or "Conclusion" in the text. Currently, the paper is rather a technique report for the authors themselves rather than new findings that the whole community can learn from, since the effect of CS2 SIT on sea ice prediction has been studied widely and comprehensively for years including CICE model (literatures can be found easily not limited to what the authors currently provided). Therefore, a paragraph with "Summary" or "Conclusion" is necessary for distinct new findings specifically in this study.

That's an excellent suggestion. We've made efforts to address this with a more specific and informative 'Summary and Conclusion' section in terms of new findings, including quantifying the seasonal performance of CICE, identifying its first-order biases, and the skill gained with a more realistic sea ice thickness initialization. We also emphasized the importance of SST and highlighted that the suitability of CICE for seasonal prediction depends on various factors, including initial conditions such as sea ice thickness, in addition to sea ice coverage, as well as oceanic and atmospheric conditions.

Regarding "2.3 Initial Conditions", the process to map CS2 SIT onto native model grid is not clear, for example, how to redistribute the mean thickness data to each category? what do the authors mean about the vicinity data to fill the North Pole, to what extent? which CS2 SIT record do you use, the daily or weekly? how the interpolation does? Bilinearly or conserved remapping?

Thank you. We have added more info to this section in the manuscript.

We utilize the monthly CS2 SIT dataset, where the mean ice thickness data is interpolated onto the CICE model grid as a single thickness category to initialize CICE. Due to missing data near the North Pole in the raw dataset, the conserved remapping method is suboptimal. Instead, we employ an interpolation method that is closer to bilinear interpolation, where the ice thickness at each model grid point is calculated as the average of all the raw data points within a radius of 2 grid spacings.

To address the challenge of missing data near the North Pole in the CS2 dataset, we adapt our approach to fill the model grid points located north of 87°N by expanding the search radius to 7 grid spacings at 87°N, and to 10 grid spacings at 89°N. This creates a more smoothly varying ice thickness field, which is important to be used as the initial condition.

Captions for Figure1 is not correct.

Thank you for catching the error in the caption. We have revised it to maintain consistency with the figure.

L119, we normally cite Zhang's paper instead of Schweiger et al., 2011 as Zhang is the main developer.

Thank you for bringing this to our attention. We concur and have replaced it by Zhang and Rothrock, 2003.

L128-L131: please rephrase the text to discuss the results quantitively.

Thanks. We have revised this to

*Fig. 1(c) shows that the modeled Antarctic SIE has a positive bias of around 20% compared to NSIDC observations at a 0.5-month lead time in all seasons. The most substantial bias occurs during austral spring, similar to the Arctic. The positive SIE bias becomes even more pronounced at a 5.5-month lead time, reaching as much as 80% of NSIDC observations during austral spring. Additionally, at a 5.5-month lead time, the model shows a ratio of annual maximum to minimum SIE that appears to be excessively large compared to the observations, with approximately a factor of 3 in the Arctic and a factor of 5 in the Antarctic.*

L134: Normally the seasonality of sea ice in the Antarctic cannot be like that, it's not about the initialization but rather a fundamental problem in the model!

We agreed. We have clarified in the text as:

*Neglecting horizontal transport in the ocean is impractical, especially in the Southern Ocean, where the Antarctic Circumpolar Current is a significant part of the global thermohaline circulation. This factor may contribute to the SIE bias, especially over longer lead times.*

Figure 2a,b: I didn't get it why Nov has such distinct small bias over lead month >2? Same in Figure3 but in April.

A good question. Identifying reasons for a significant bias seems more straightforward than pinpointing the causes of a small bias, as small bias can arise either for valid reasons or due to errors from different sources canceling each other out, resulting in a minimal overall bias.

We plotted the simulated SIE in the control experiments, a 5-year average, against that from NSIDC in Arctic and Antarctic, see below, where * marks the beginning of the 12-month integrations. Clearly, the modeled SIE has a positive bias at almost all lead time, with the minimum bias in November in the Arctic, and in April in Antarctica. One approach to unravel this would be to examine the SIE tendencies from thermodynamics and dynamics. However, without observational data for comparison, we still won't be able to explain this phenomenon. Our intention is to contrast these quantities with those derived from coupled model experiments in future experiments in the hope of gaining insights.

On the other hand, higher skills in fall are also seen in other studies, e.g., Peterson et al. 2015, Martin et al. 2023. We have added references to these studies.

[Figure]

L187: With respect to tendencies arising from the thermodynamics and dynamics, for readers don't use CICE will never know what that means! A description on which terms the two terms account for is necessary.

Thanks for your suggestion. This part was added as a response to a previous reviewer. Now we added more description:

*The origins of SIE change can be categorized into two tendencies, originating from thermodynamics and dynamics, respectively. Same is true for SIV tendencies.*

---

## Author Response (AR1)

**Reviewer #1**

We thank the reviewer for their careful reading of the manuscript and constructive remarks, which helped improve the quality of the manuscript. We have addressed all the recommendations, see details below (reviewer's comments in black, our replies in blue).

The Los Alamos sea ice model (CICE) is tested in standalone mode in the study and the performance on seasonal sea ice prediction is examined. The paper is well written and organized with clear logics that readers can readily follow. The results present now is informative. However, one more aspect can be considered to further improve the manuscript. My comments are generally listed below:

I would recommend add "Summary" or "Conclusion" in the text. Currently, the paper is rather a technique report for the authors themselves rather than new findings that the whole community can learn from, since the effect of CS2 SIT on sea ice prediction has been studied widely and comprehensively for years including CICE model (literatures can be found easily not limited to what the authors currently provided). Therefore, a paragraph with "Summary" or "Conclusion" is necessary for distinct new findings specifically in this study.

That's an excellent suggestion. We've made efforts to address this with a more specific and informative 'Summary and Conclusion' section in terms of new findings, including quantifying the seasonal performance of CICE, identifying its first-order biases, and the skill gained with a more realistic sea ice thickness initialization. We also emphasized the importance of SST and highlighted that the suitability of CICE for seasonal prediction depends on various factors, including initial conditions such as sea ice thickness, in addition to sea ice coverage, as well as oceanic and atmospheric conditions.

Regarding "2.3 Initial Conditions", the process to map CS2 SIT onto native model grid is not clear, for example, how to redistribute the mean thickness data to each category? what do the authors mean about the vicinity data to fill the North Pole, to what extent? which CS2 SIT record do you use, the daily or weekly? how the interpolation does? Bilinearly or conserved remapping?

Thank you. We have added the paragraph below to this section in the manuscript:

*We utilize the monthly CryoSat-2 dataset, where the mean ice thickness data is interpolated onto the CICE model grid as a single thickness category to initialize CICE. We employ an interpolation method that is close to bilinear interpolation, where the ice thickness at each model grid point is calculated as an average of all the raw data points within a radius of 2 grid spacings.*

*It is important to note that the original CryoSat-2 dataset does not provide ice thickness data in the immediate vicinity of the North Pole. To address this limitation, we estimated ice thickness values at grid points within this region from the surrounding ice thickness data using bilinear interpolation with a bigger search radius (7 grid spacings at 87°N and 10 grid spacings at 89°N). This interpolation process allowed us to fill the data gap near the North Pole with a smoothly varying ice thickness field.*

Captions for Figure1 is not correct.

Thank you for catching the error in the caption. We have revised it to maintain consistency with the figure.

L119, we normally cite Zhang's paper instead of Schweiger et al., 2011 as Zhang is the main developer.

Thank you for bringing this to our attention. We concur and have replaced it by Zhang and Rothrock, 2003.

L128-L131: please rephrase the text to discuss the results quantitively.

Thanks. We have revised this to:

*Fig. 1(c) shows that the modeled Antarctic SIE has a positive bias relative to NSIDC observations across all seasons at a 0.5-month lead time, with the biggest bias occurring during austral spring, similar to the Arctic. A larger positive SIE bias is seen during austral spring at a 5.5-month lead time, resulting in an annual range of SIE that is excessively large compared to observations, approximately a factor of 5 in the Antarctic and a factor of 3 in the Arctic.*

L134: Normally the seasonality of sea ice in the Antarctic cannot be like that, it's not about the initialization but rather a fundamental problem in the model!

We agreed. We have clarified in the text as:

*The most pronounced positive bias in SIE occurs during winter in both the Arctic and Antarctic regions. This issue is closely linked to the simplistic representation of ocean dynamics within the utilized mixed-layer ocean model in the standalone configuration. Here, a one-dimensional column model is employed, without accounting for horizontal ocean transport. This bias is most evident in the Labrador Sea and Bering Sea near the Arctic, as well as in the Southern Ocean surrounding the Antarctic. These regions are particularly influenced by the North Atlantic Deep Water and the Antarctic Circumpolar Current, both integral components of the global thermohaline circulation.*

*Consequently, neglecting oceanic heat transport is likely to lead to a positive bias in SIE, particularly over extended lead times.*

Figure 2a,b: I didn't get it why Nov has such distinct small bias over lead month >2? Same in Figure3 but in April.

A good question. Identifying reasons for a significant bias seems more straightforward than pinpointing the causes of a small bias, as small bias can arise either for valid reasons or due to errors from different sources canceling each other out, resulting in a minimal overall bias.

We plotted the simulated SIE in the control experiments, a 5-year average, against that from NSIDC in Arctic and Antarctic, see below, where * marks the beginning of the 12-month integrations. Clearly, the modeled SIE has a positive bias at almost all lead time, with the minimum bias in November in the Arctic, and in April in Antarctica. One approach to unravel this would be to examine the SIE tendencies from thermodynamics and dynamics. However, without observational data for comparison, we still won't be able to explain this phenomenon. Our intention is to contrast these quantities with those derived from coupled model experiments in future experiments in the hope of gaining insights.

On the other hand, higher skills in fall are also seen in other studies, e.g., Peterson et al. 2015, Martin et al. 2023. We have added references to these studies:

*Considering the potential for the RMSE in SIE to be larger during winter due to the extended ice edge and greater ice extent compared to other seasons, it is informative to compare the RMSE with the interannual variabilities of SIE. The interannual variabilities of SIE, measured by the standard deviation, and shown in Fig. 1(a), do not show a clear seasonal dependency at a 6-month lead time, consistent with the NSIDC observation. Nevertheless, the RMSE in the Arctic SIE, shown in Fig. 2(a), reveals a distinct seasonal cycle. The RMSE and the standard deviation for Arctic SIE at a 6-month lead time are comparable in late fall, indicating a higher forecast skill during this season. In contrast, the RMSE substantially exceeds the interannual variabilities in late winter. The decline in forecast skills for late winter is consistent with findings in Peterson et al. (2015); Martin et al. (2023).*

[Figure]

L187: With respect to tendencies arising from the thermodynamics and dynamics, for readers don't use CICE will never know what that means! A description on which terms the two terms account for is necessary.

Thanks for your suggestion. This part was added as a response to a previous reviewer. Now we added more description:

*The origins of SIE change can be categorized into two tendencies, originating from thermodynamics and dynamics, respectively. The same is true for SIV tendencies.*

**Reviewer #2**

We thank the reviewer for their careful reading of the manuscript and constructive remarks, which helped improve the quality of the manuscript. We have addressed all the recommendations, see details below (reviewer's comments in black, our replies in blue).

**Manuscript Synopsis**

This manuscript evaluates the usage of the CICE sea ice model for seasonal forecasting in NOAA's Unified Forecast System by examining 12 month simulations of CICE driven by reanalysis atmospheric forcing and an ocean 1-D mixed layer model. Using root mean square errors in integrated quantities, the authors evaluate hemispheric wide errors in sea ice extent and sea ice volume - but not spatial errors.

My evaluation of this manuscript is to reconsider after major revisions. These major revisions (major comments 1, 2 and 4 below) can mostly be dealt with by the supply of additional information either in text or in response. Major comment 3 is a recommendation which could enhance the manuscripts applicability, but is not critical to its publishability.

The most pressing issue, which may stem solely from my lack of understanding of running CICE in stand-alone mode and comes down to a lack of details with respect to the oceanic boundary condition (Section 2.2). While I understand the authors usage of the mixed layer ocean model for the sea ice thermodynamics, it does not explain what was used for bottom boundary dynamical conditions – and leads me to believe there are none, or more precisely, it assumes an unmoving ocean. Perhaps this could be resolved with some rather simple re-working of the model introduction (i.e. more to Section 2.2 then just the thermodynamic lower B.C.). As the location of the thickest ice is predominantly set by dynamic processes (driven against the Canadian Archipelago and northern Greenland) – and the location of the thickest ice in the model setup is not modelled very well (in the Beaufort Sea and central Canada Basin), this suggests a possible deficit in dynamical tendencies that requires a better articulation of the dynamical lower boundary conditions. [Scientific Quality: Methods not adequately explained.]

Thank you for your comment on this. Indeed, we assumed the ocean to be stationary, and there was no specified dynamical condition for the bottom boundary. We have now included this clarification in the manuscript:

*The most pronounced positive bias in SIE occurs during winter in both the Arctic and Antarctic regions. This issue is closely linked to the simplistic representation of ocean dynamics within the utilized mixed-layer ocean model in the standalone configuration. Here, a one-dimensional column model is employed, without accounting for horizontal ocean transport. This bias is most evident in the Labrador Sea and Bering Sea near the Arctic, as well as in the Southern Ocean surrounding the Antarctic. These regions are particularly influenced by the North Atlantic Deep Water and the Antarctic Circumpolar Current, both integral components of the global thermohaline circulation. Consequently, neglecting oceanic heat transport is likely to lead to a positive bias in SIE, particularly over extended lead times.*

A 2nd major omission of the manuscript is the mistaken impression implied in the introduction (ll. 37-41; and granted it is not stated explicitly) that CICE has mainly been used for climate simulations. This is far from the truth, as CICE is already in use in many operational seasonal, sub-seasonal and NWP systems, which I detail further below. [Originality: Manuscript does not adequately present current scientific Understanding.]

Thank you for pointing this out. We have now updated this in the manuscript:

*CICE, originally developed for long-term climate research, has been used in seasonal prediction applications with success and challenges, for example, in the Global Seasonal forecast systems at the UK Met Office and the Canadian Seasonal to Interannual Prediction System (CanSIPSv2) (Arribas et al., 2011; MacLachlan et al., 2015; Lin et al., 2020). In addition, Martin et al (2023) attributed the enhanced skill in CanSIPSv2 compared to the previous version to an improved sea ice initialization procedure.*

Lastly, while the skill metrics used in the manuscript are acceptable – but very climate oriented – they can be enhanced quite easily to assess integrated errors as opposed to errors in integrated quantities (IIEE), simply by changing the order in which the operations are performed. This point I would be willing to give the authors some leeway with. The regional results (Section 3.4) do address this issue somewhat – although Figure 11 is not correct, but Figure 12 does not suggest any cancellation of error (only deficits in sea ice). Nevertheless it could highlight further errors not elicited by Figures 2 and 3. [Scientific Quality: Validation methods could be improved. Significance: Application of results could be improved through enhanced spatial Information.]

Could you please provide more information to the error in Fig. 11? Figs. 11 and 12 are identical, differing only in the choice of initial months. Fig. 11 is for experiments starting on April 1, close to the annual maximum SIE, and Fig. 12 is for experiments starting on Oct. 1, close to the annual minimum SIE. Both sets of experiments, regardless of the initial month, reveal a positive bias in both SIE and SIV in the BKG Seas and Baffin Bay. We attribute this bias to the model's lack of northward oceanic heat transport.

I appreciate that this manuscript may already have been through at least one round of review and revision, so I hope my suggestions do not pose too onerous a task.

My recommendation is Reconsider after Major Revision.

**Major Comments**
1. Section 2.2: No explanation is given for lower (ocean) dynamic boundary condition. While the authors dedicate a sub-section (Section 2.2) to the ocean boundary conditions, this only explains the lower boundary condition for Sea Surface Temperature (SST) – which is chosen to be a mixed layer 1-D ocean – and would only effect the model thermodynamics. No explanation for specification of the dynamical boundary conditions, or more precisely, the ocean surface currents is given. Presumably, this would have a large effect on the sea ice dynamics, which it is impossible to make informed decisions regarding without further information.

The one-dimensional mixed layer ocean model has a thickness of 20 m and is stationary, solely designed for sea ice thermodynamics. The omission of horizontal advection in the ocean from this setup has notable implications for the results, as highlighted in the manuscript.

2. ll. 37-41: The introductory paragraph gives the false impression that the CICE sea ice model is primarily used for climate simulations (Note: I do not deny that the model was initial constructed for this purpose), implicitly implying that its introduction as the sea ice component for NOAA's Unified Forecasting System is a novel usage. The exact phrasing used (ll 39-40) "its suitability for seasonal forecasting needs to be assessed." I do agree, or at least do not disagree with that statement, however, some credit through citation is deserving to the multitude of operational (and quasi-operational) systems currently in use for seasonal forecasting use throughout the world for over a decade (Note: many of these, by necessity, are self-serving):
( 1) UK Met Office GloSea4/5 system (https://doi.org/10.1175/2010MWR3615.1, https://doi.org/10.1007/s00382-014-2190-9, https://doi.org/10.1002/qj.2396)

( 2) Korea Met Agency version of GloSea5 (https://doi.org/10.5194/tc-2018-217)
( 3) ASSESS-S1/2, Australia Bureau of Meteorology version of GloSea5 (https://www.publish.
csiro.au/es/ES17009, https://www.publish.csiro.au/ES/ES22026).
( 4) CanSIPSv2 GEM-NEMO-CICE component (https://doi.org/10.1175/WAF-D-19-0259.1, https:
//doi.org/10.1175/WAF-D-22-0193.1, https://www.tandfonline.com/doi/pdf/10.1080/07055900.
2023.2252387

6 of 11 dynamical contributors to the ARCUS July 2023 sea ice outlook
(https://www.arcus.org/sipn/sea-ice-outlook/2023/july (5 listed below as UK Met Office is one of 6)

( 5) RASM/NPS (https://doi.org/10.5194/gmd-11-4817-2018)
( 6) ArcIOAM, National Marine Environmental Forecasting Center, China (https://doi.org/10.
5194/gmd-14-1101-2021)
( 7) FIO-ESMv1.0, Qingdao, China (https://doi.org/10.3389/fmars.2020.00504
( 8) FGOALS-f2 V1.3, Institute of Atmospheric Physics, China (https://doi.org/10.1029/2019MS002012)
( 9) Unified Forecast System, NOAA (https://doi.org/10.1029/2022GL102392). Note: This is the
prototype system being evaluated in this manuscript. and a final system via a google search for CICE seasonal
forecasts
(10) SLAV/INMIO/CICE, Marchuk Institute of Numerical Mathematics / Shirshov Institute of Oceanology /
Hydrometeorological Centre of Russia (https://doi.org/10.1515/rnam-2018-0028)
to which I will also add shorter range S2S, monthly and short range (< 10day) systems:
( 11) Global Ensemble Prediction System, Environment and Climate Change Canada (https://doi.
org/10.1002/qj.4340) (S2S/monthly/extended)
( 12) Global Ice Ocean Prediction System / Regional Ice Ocean Prediction System, Environment and
Climate Change Canada (short range) (https://doi.org/10.1002/qj.2555, https://doi.org/
10.1175/MWR-D-17-0157.1, https://doi.org/10.5194/gmd-14-1445-2021)
2
( 13) Forecasting Ocean Assimilation Model (FOAM), UK Met O ce (https://doi.org/10.5194/
gmd-7-2613-2014) (short range)
( 14) Prototype UK Met Office Coupled System for NWP (https://doi.org/10.1175/WAF-D-20-0035.
1) (short range)

While I would not expect (the list was longer than even I had initially assumed!) the authors to
cite each and every one of these, it would still be appropriate to underline the usage of the CICE
model in existing operational systems – and highlight and reference their results against the sea ice
predictability of some of these earlier systems, particularly when an assessment of sea ice performance
has been undertaken. See point 4 for one such possible connection.

Thank you so much for bringing this to our attention and for providing a comprehensive list of references. We have
revised the manuscript accordingly, as mentioned earlier.

3. RMSE skill measure: While the RMSE quantification of error for the hemispheric domain is ade-
quate, it is also relatively non-standard. More usual to be found in seasonal papers is anomaly cor-
relation (https://doi.org/10.1002/grl.50129, https://doi.org/10.1007/s00382-014-2190-9),
which eliminates bias. However, more modern skill estimates account for both the area of the sea
ice extent along with its position, through skill measures like Integrated Ice Edge Error (https:
//doi.org/10.1002/2015GL067232). Implementation of this would be simple enough. All you need
do is commute the order in which the area and rmse error operations are performed. [I.e. calculate
the square error of ice existence $(M - O)^2$, where M is modelled ice > 0.15 concentration and O is

observed ice > 0.15 concentration in any grid cell, and then perform your summation. Note: Since (M - 0) is 1/0 it does not matter whether you square or take absolute value. Taking the square easily allows you to generalize for an ensemble, where M is replaced with P, the fraction of ensemble members with ice (https://doi.org/10.1002/qj.3242). You could also consider replacing the RMSE of total ice volume with the integrated square error of grid cell ice volume (it is preferable to add square error, not root mean square error). The latter will then give you a double penalty for having ice volume in the Beaufort Sea, but little over the Canadian Archipelago and Greenland. I will not insist on the authors doing this, but it could enhance the applicability of their results.

We appreciate your suggestion regarding IIEE and the accompanying reference. Indeed, the computation of IIEE and RMSE diverges in the order of summation operations. Unfortunately, the existing NSIDC data available to us provides only hemispheric totals. The calculation of IIEE requires data at each grid point, which is currently impractical for us due to time constraints.

While the current circumstances hinder our ability to calculate IIEE, we acknowledge its significance as a valuable metric. We are committed to exploring the inclusion of IIEE in future studies when the availability of more data permits. We have included this in the manuscript:

*Goessling et al. (2016) introduced a useful metric for SIE known as the 'Integrated Ice Edge Error', which calculates the integral of all mismatched areas between modeled and observed SIE. This metric provides the advantage of distinguishing between absolute extent error (AEE) and misplacement error (ME). In our study, since AEE predominates over ME, as shown later in Fig. 5, we utilize RMSE and bias to evaluate SIE. This approach allows for a direct comparison with the widely-used Arctic and Antarctic SIE data from NSIDC.*

4. Although it is also a characteristic of the IEEE as well (so that will not solve this problem), the RMSE of an area integrated quantity will be inherently larger when that integrated quantity is larger. Thus it may be natural for the RMSE in February and March to be large solely because the ice extent is large during those periods. It would be a more accurate assessment of whether the ice area predictability is better or worse by comparing the RMSE with the interannual variability for that time of year. That being said, deterioration of predictability seems also to occur for mid to late winter in many other assessments of sea ice predictability through correlation skill assessments (https://doi.org/10.1002/grl.50129, https://doi.org/10.1007/s00382-014-2190-9, https://doi.org/10.1175/WAF-D-22-0193.1). Perhaps the authors can comment on this – and cite previous seasonal sea ice skill Assessments.

It is a good point. We compared the RMSE in the Arctic SIE with the interannual variabilities of the Arctic SIE, see below. The two are very similar in late fall, but the RMSE in late winter is much larger than the STD of the Arctic SIE.

[Figure]

We have added this paragraph in the manuscript:

*Considering the potential for the RMSE in SIE to be larger during winter due to the extended ice edge and greater ice extent compared to other seasons, it is informative to compare the RMSE with the interannual variabilities of SIE. The interannual variabilities of SIE, measured by the standard deviation, and shown in Fig. 1(a), do not show a clear seasonal dependency at a 6-month lead time, consistent with the NSIDC observation. Nevertheless, the RMSE in the Arctic SIE, shown in Fig. 2(a), reveals a distinct seasonal cycle. The RMSE and the standard deviation for Arctic SIE at a 6-month lead time are comparable in late fall, indicating a higher forecast skill during this season. In contrast, the RMSE substantially exceeds the interannual variabilities in late winter. The decline in forecast skills for late winter is consistent with findings in Peterson et al. (2015); Martin et al. (2023).*

5. Figure 11: Figure 11 is an error/omission. Figure 11 is identical, save for 6 month offset x-axis to the correctly attributed Figure 12.

Could you please provide more information to the error in Fig. 11? Figs. 11 and 12 are similar, differing only in the choice of initial months. Fig. 11 is for experiments starting on April 1, close to the annual maximum SIE, and Fig. 12 is for experiments starting on Oct. 1, close to the annual minimum SIE. Both sets of experiments, regardless of the initial month, reveal a positive bias in both SIE and SIV in the BKG Seas and Baffin Bay. We attribute this bias to the model's lack of northward oceanic heat transport.

**Minor Comments**
1. Section 2.1: You should emphasize that you are forcing the sea ice integration with "0-hour lead" reanalysis forcing. In other words, you are not performing a true seasonal forecast, where the forcing is also of long lead time.

Thank you for pointing this out. We have added that "The *prescribed time-varying* atmospheric boundary forcings used in this study are derived from the 6-hourly archives obtained from CFSR reanalysis".

2. Section 2.2: The above point then begs the question as to why "0-hour lead" SST forcing does not lead to a better ice concentration (not necessary thickness) integration as described in Guemas et al (2014; http://dx.doi.org/10.1007/s00382-014-2095-7). Perhaps it did – the explained reason for abandoning due to it "result(ing) in an unrealistic increase in basal melt," was for reasons of unrealistic thermodynamics, it may still have resulted in a more accurate sea ice concentration integration – likely at the cost of a more unrealistic sea ice thickness integration. Perhaps the authors can expand their Explanation.

Appreciate the reference provided. The performance degradation of CICE was unfortunately influenced by a positive bias in SST from the CFSR. It's reassuring to note that the ocean temperature in ORAS4 exhibited strong performance, as highlighted in Guemas et al. (2014), surpassing that of CFSR. We have incorporated the reference to Guemas et al. (2014) into the manuscript, explicitly acknowledging that a minor yet persistent positive bias in CFSR SST data led to the unrealistic basal melt seen in the initial experiment. Consequently, we had to abandon the original approach. We have revised this paragraph as following:

*CICE requires information on the sea surface temperature (SST), which can be either prescribed directly or generated inline using the built-in mixed layer ocean model within CICE. Initially, the CFSR SST data was prescribed to drive the CICE model. However, it was found that this approach led to an unrealistically large basal melt, primarily attributed to a small yet persistent positive bias in the CFSR SST data, despite the success in Guemas et al. (2014), where ocean temperature and salinity are nudged towards the NEMOVAR ORAS4 ocean reanalysis (Mogensen et al., 2011). To address this issue and ensure consistency between the SST and the ice state, an alternative method was adopted, i.e., to use the mixed layer ocean model within CICE to prognose the SST. This means that the CICE model itself generates the SST information based on the physical processes happening within the mixed layer of the ocean. By employing this approach, the SST and ice state remain consistent, allowing for a more reliable representation of the interaction between the sea ice and the ocean in the model simulations than the earlier attempt. However, it's important to note that the mixed layer ocean model used in this approach is a simplified one dimensional stationary model that does not include horizontal advection in the ocean. This limitation does impact the model results, as will be demonstrated later.*

3. Section 2.3: It is very important you explicitly specify you initialized to the Cryosat-2/SMOS dataset (Ricker et al, 2014, https://doi.org/10.5194/tc-11-1607-2017). Otherwise readers will be confused on how you initialized sea ice with thickness less than 1m.

Unfortunately, we utilized the raw CryoSat-2 data without knowledge of CryoSat-2/SMOS as mentioned in Ricker et al. 2017. Consequently, there are large uncertainties in sea ice thickness over thin ice regimes. We have acknowledged this limitation in the manuscript, and it is an aspect we intend to enhance in future experiments. We have added this in the manuscript:

*Furthermore, the CryoSat-2 dataset itself may also contribute to the biases, as relative uncertainties are high over thin ice regimes, given that sea ice thickness is determined by the ice surface above the sea level (Ricker et al., 2017).*

**Minor Presentation Comments**
1. Perhaps this is pedantic, but the units should really be on the colour bar (Figures 2, 3, 4, 5, 6, 8, 9), or on the y-axis (Figures 7, 11, 12) if possible, and not just (could be additionally) in the figure title.
Note: Figure 9, lacks units completely – although fairly obviously ºC/K. The latter at least needs to be corrected.

Thanks for your suggestion. We are able to add units to all colorbars in the manuscript successfully.

**Editor**

- Add a distinct Section "Conclusions" summarizing the main benefits from your study.

Thanks. A conclusion is added to highlight new findings, including quantifying the skill gained with a more realistic sea ice thickness initialization. We also emphasize the importance of SST and underscore that the suitability of CICE for seasonal prediction depends on various factors, including initial conditions such as sea ice thickness, in addition to sea ice coverage, as well as oceanic and atmospheric conditions.

- Describe your model setup in more detail and make clear what are the constraints of you setup (simple mixed-layer ocean model) and what you can still conclude inspite your setup.

It is unfortunate that a simplified one dimensional stationary model had to be used in this study that did not include horizontal advection in the ocean. Despite this limitation, the model is capable of delivering a decent SIE forecast, particularly during the warm season at all lead times up to 12 months. We have added additional discussion to address this limitation.

- Ideal skill measure and impact of variability (Point 3 and 4 of Ref. 2).

We have addressed both issues in the revised manuscript.

---

## Referee Report (RR1)

**2nd Review of: Suitability of CICE Sea Ice Model for Seasonal Prediction and Positive Impact of CryoSat-2 Ice Thickness Initialization**

by Shan Sun and Amy Solomon

April 2, 2024

**Manuscript Synopsis**

This manuscript evaluates the usage of the CICE sea ice model for seasonal forecasting in NOAA's Unified Forecast System by examining 12 month simulations of CICE driven by reanalysis atmospheric forcing and an ocean 1-D mixed layer model. Using root mean square errors in integrated quantities, the authors evaluate hemispheric and regional errors in sea ice extent and volume.

**My evaluation of this manuscript is to accept with minor technical revisions.** The authors have commented on and taken into account in the text of the manuscript the major issues I had with regards to an earlier version of the manuscript. I now find the manuscript suitable for publication.

**My recommendation is Accept with minor technical revisions.**

**Major Comments**

1. My previous major comments have been addressed, and I thank the authors for taking them into account.

2. I do have some hesitation still with regards to the missing ocean dynamics with the use of an ocean mixed layer. In particular, the authors seem to acknowledge the only missing component of the ocean dynamics is missing heat transport by the ocean. While this is certainly important for the Barents and Kara Seas, an equally missing component is sea ice transport via ocean currents – as opposed to wind forcing, which is present in their simulations. Oceanic advection undoubtedly plays a role in their forecasting results in the Labrador Sea and East Greenland regions, but also likely has a large influence in the Beaufort and Chukchi seas – although here, the elimination of the sea ice thickness bias from CRSR initialization is by and large the most important driver of increased skill.

3. Instead of listing further major comments, I have added a (non-compulsary) list of additional comments that I and other readers might have considering the manuscript. I am not suggesting the authors need address them, but rather they take note of them for future reference.

**Minor Technical Comments**

1. Abstract. In the abstract, can you be more specific then "the model's Arctic sea ice thickness has a positive bias"? Perhaps

   the model's Arctic sea ice initial conditions have too thick ice in the Beaufort Sea, leading to too large ice extents in the Arctic at 6 month lead forecasts and when observable, errors in all leadtime forecasts of ice volume.

*I realize the situation with regards to forecasts of ice volume are perhaps more convoluted, but I was limiting my suggestion to a single sentence.*

2. ll 40-41. The inserted (thank you) citations for the existing seasonal forecasts that use CICE are slightly confusing. Could you please separately cite the MetOffice GloSea system (specifically say the GloSea system) and the Canadian CanSIPS systems? It is also in this way perhaps fairer to the MetOffice system which pre-dates CanSIPS use of CICE by almost a decade. E.G.

   $\cdots$ for example, in the UK Met Office Global Seasonal (GloSea) forecast system (Arribas et al., 2011; MacLachlan et al., 2015) and in the Canadian Seasonal to Interannual Prediction System (CanSIPSv2) (Lin et al., 2020).

3. ll. 119-120. The statement "Lead times not ending in .5 are rounded up to the nearest integer month for simplicity (i.e., 11.5-month lead time is rounded up to 12-month lead time)" seems contradictory to me (i.e. by your first statement lead times ending in 0.5 should should not be rounded at all). I'm not sure the whole statement is necessary, and could simply be removed.

4. Figure 7. The dotted (CS2_IC) lines seem like dashed lines to me?

**Additional Comments to Consider**

I am not asking for any action on any of these listed items. However, you may wish to consider how these are presented in the future.

1. ll. 53-59. While a complete set of CICE namelist options/values would be outside the scope of a *Model Setup* section (but might be part of open data supplementary material), one common aspect of CICE model setup beyond the EVP, ice thickness categories, and ice/snow thermodynamic layers is whether or not a meltpond parameterization scheme is used.

2. l. 86. The mixed layer in the Arctic varies considerably with season (Uotila et al. An assessment of ten ocean reanalyses in the polar regions. Clim Dyn 52, 1613?1650 (2019). https://doi.org/10.1007/s00382-018-4242-z; and observational citations within). I have no knowledge in terms of the capabilities of the mixed layer ocean scheme in CICE, but is a constant 20m choice for mixed layer depth a reasonable and realisitic choice. This might have 2nd order implications for the biases you observe in SST (Figure 9) – although the 1st order biases are obviously oceanic heat transport.

3. ll 99-100. Assigning zero values to areas where CryoSat-2 observes no sea ice thickness – which might be anywhere with sea ice thickness below 1m – would seemingly reduce your initial condition sea ice concentrations and sea ice extent (although Figure 4 top row shows no evidence for this). Might this be worth expanding on? Are there areas in CS2_IC where sea ice concentration was removed relative to CNTR IC?

4. ll 185-189. I find the oceanic heat transport explanation for positive bias in sea ice concentration in the Labrador Sea (and elsewhere) unsatisfying. An alternative, or likely easier explanation might be found in ocean current transport of ice not incorporated in the mixed layer ocean model. The Labrador Sea and along the east coast of Greenland are regions with a southward ocean current that moves and disperses sea ice southward (where it melts). A lack of ice transport could very easily explain the positive bias of sea ice concentration here – and perhaps in the southern ocean as well – although ice transport through atmospheric forcing plays a large role there. Overall, while the lack of ocean heat transport is obviously important – especially in the Barents and Kara Sea – but other than there, a lack of sea ice transport by ocean currents is likely to be just as big a missing factor imposed by the mixed layer representation of the ocean. And then there would also be the role played by the imposed depth of the mixed layer itself.

5. Figure 4 and spatial biases discussed in (previous) sub-section 3.1: Although I understand the plotting of ice concentration fields for October 6.5 month forecasts: It matches your ability to show similar plots with observation for ice thickness in Figure 5. However it is the April initialized forecast lead with the

smallest bias (Figure 1a; Figure 11; are AMSR2 observations really only available Oct-Apr?), which hides significant system biases, particularly in the Barents and Kara Seas not mentioned explicitly until sub-section 3.4. Some better co-referencing of spatial biases with hemispheric biases and a better coherence with regards to which start dates to concentration on could have been achieved in the text [and I should have taken more careful note in the first revision of this paper].

6. Figure 9. Due to some of the lack of connections made in my previous point the plot of SST biases (Figure 9) lacks cohesion with the rest of the manuscript. The influence (or lack there-of) of oceanic heat transport is obvious from the figure, but then it is hard to then compare this with the sea ice results already shown – or to possibly link the contribution of other system biases (CFSR forcing / depth of mixed layer) to both SST biases and sea ice biases.

---

## Author Response (AR2)

We thank the reviewer for their diligent review and insightful feedback on our revised manuscript, which have significantly enhanced the quality of our manuscript. We have addressed all the recommendations, see details below (reviewer's comments in black, our replies in blue).

This manuscript evaluates the usage of the CICE sea ice model for seasonal forecasting in NOAA's Unified Forecast System by examining 12 month simulations of CICE driven by reanalysis atmospheric forcing and an ocean 1-D mixed layer model. Using root mean square errors in integrated quantities, the authors evaluate hemispheric and regional errors in sea ice extent and volume. My evaluation of this manuscript is to accept with minor technical revisions. The authors have commented on and taken into account in the text of the manuscript the major issues I had with regards to an earlier version of the manuscript. I now find the manuscript suitable for publication.

My recommendation is Accept with minor technical revisions.

Major Comments
1. My previous major comments have been addressed, and I thank the authors for taking them into account.

Thank you.

2. I do have some hesitation still with regards to the missing ocean dynamics with the use of an ocean mixed layer. In particular, the authors seem to acknowledge the only missing component of the ocean dynamics is missing heat transport by the ocean. While this is certainly important for the Barents and Kara Seas, an equally missing component is sea ice transport via ocean currents – as opposed to wind forcing, which is present in their simulations. Oceanic advection undoubtedly plays a role in their forecasting results in the Labrador Sea and East Greenland regions, but also likely has a large influence in the Beaufort and Chukchi seas – although here, the elimination of the sea ice thickness bias from CRSR initialization is by and large the most important driver of increased skill.

Thanks for pointing this out. We have added this to the summary:

*This positive bias in SIT is likely due to the absence of oceanic currents in the column mixed layer ocean model,which, although having a lesser impact compared to wind forcing, still plays a role in sea ice export.*
*Additionally, the cold bias in the modeled SST near the marginal ice zone may contribute to the positive bias in SIT. The cold bias is likely a consequence of neglecting poleward oceanic heat transport within the simplified ocean model used in the experiments.*

3. Instead of listing further major comments, I have added a (non-compulsary) list of additional comments that I and other readers might have considering the manuscript. I am not suggesting the authors need address them, but rather they take note of them for future reference.

We appreciate this list. We addressed them as much as we can here and will use it for future reference.

Minor Technical Comments
1. Abstract. In the abstract, can you be more specific then "the model's Arctic sea ice thickness has a positive bias"? Perhaps the model's Arctic sea ice initial conditions have too thick ice in the Beaufort Sea, leading to too large ice extents in the Arctic at 6 month lead forecasts and when observable, errors in all leadtime forecasts of ice volume.

I realize the situation with regards to forecasts of ice volume are perhaps more convoluted, but I was limiting my suggestion to a single sentence.

Thank you. We have revise this in the abstract:

*The model's initial conditions have too thick ice in the Beaufort Sea, resulting in excessive ice extent*
*in the Arctic at 6-month lead forecasts and errors in ice volume at all lead times when compared to available observations.*

2. ll 40-41. The inserted (thank you) citations for the existing seasonal forecasts that use CICE are slightly
confusing. Could you please separately cite the MetOffice GloSea system (specifically say the GloSea
system) and the Canadian CanSIPS systems? It is also in this way perhaps fairer to the MetOffice system which pre-dates CanSIPS use of CICE by almost a decade. E.G. · · · for example, in the UK Met Office Global Seasonal (GloSea) forecast system (Arribas et al., 2011; MacLachlan et al., 2015) and in the Canadian Seasonal to Interannual Prediction System (CanSIPSv2) (Lin et al., 2020).

Thank you for your suggestion; we've incorporated it into the revision. During our work with the CICE5 source code, we noticed the valuable contributions made by the Met Office.

3. ll. 119-120. The statement "Lead times not ending in .5 are rounded up to the nearest integer month
for simplicity (i.e., 11.5-month lead time is rounded up to 12-month lead time)" seems contradictory
to me (i.e. by your first statement lead times ending in 0.5 should should not be rounded at all). I'm
not sure the whole statement is necessary, and could simply be removed.

Thank you. We have removed this. We added a footnote of "*Rounded up to the nearest integer month for simplicity*".

4. Figure 7. The dotted (CS2 IC) lines seem like dashed lines to me?

Thank you: they are indeed dashed lines. We changed "dotted" to "dashed" in the manuscript.

Additional Comments to Consider

I am not asking for any action on any of these listed items. However, you may wish to consider how these are presented in the future.

1. ll. 53-59. While a complete set of CICE namelist options/values would be outside the scope of a Model Setup section (but might be part of open data supplementary material), one common aspect of CICE model setup beyond the EVP, ice thickness categories, and ice/snow thermodynamic layers is whether or not a meltpond parameterization scheme is used.

Thank you. We have added in the manuscript that no melt pond parameterization scheme is used in this study.

2. l. 86. The mixed layer in the Arctic varies considerably with season (Uotila et al. An assessment of ten ocean reanalyses in the polar regions. Clim Dyn 52, 1613?1650 (2019). https://doi.org/10.1007/s00382-018-4242-z; and observational citations within). I have no knowledge in terms of the capabilities of the mixed layer ocean scheme in CICE, but is a constant 20m choice for mixed layer depth a reasonable and realisitic choice. This might have 2nd order implications for the biases you observe in SST (Figure 9) – although the 1st order biases are obviously oceanic heat transport.

Thank you for the reference. We have added in the manuscript:

*While this choice is generally a reasonable approximation, it may not adequately represent the actual conditions in certain regions. Specifically, this fixed depth may be too shallow in the Amundsen and Makarov Basins during winter, as well as within the Antarctic Circumpolar Current, when compared to observations as shown in Uotila et al. (2019).*

3. ll 99-100. Assigning zero values to areas where CryoSat-2 observes no sea ice thickness – which might be anywhere with sea ice thickness below 1m – would seemingly reduce your initial condition sea ice concentrations and sea ice extent (although Figure 4 top row shows no evidence for this). Might this be worth expanding on? Are there areas in CS2 IC where sea ice concentration was removed relative to CNTR IC?

Thank you. Ricker et al. (2017) pointed out the uncertainty in the CryoSat-2 dataset regarding thin ice. However, this uncertainty in the thin ice region does not invariably result in a zero ice thickness. In the CS2_IC case, ice coverage is only reduced to zero for points with precisely zero ice thickness. Consequently, only a few points fall into this category, as shown in the sea ice concentration difference plot below.

Initial Sea Ice Concentration Difference (%) CS2_IC minus Ctrl

[Figure]

4. ll 185-189. I find the oceanic heat transport explanation for positive bias in sea ice concentration in the Labrador Sea (and elsewhere) unsatisfying. An alternative, or likely easier explanation might be found in ocean current transport of ice not incorporated in the mixed layer ocean model. The Labrador Sea and along the east coast of Greenland are regions with a southward ocean current that moves and disperses sea ice southward (where it melts). A lack of ice transport could very easily explain the positive bias of sea ice concentration here – and perhaps in the southern ocean as well – although ice transport through atmospheric forcing plays a large role there. Overall, while the lack of ocean heat transport is obviously important – especially in the Barents and Kara Sea – but other than there, a lack of sea ice transport by ocean currents is likely to be just as big a missing factor imposed by the mixed layer representation of the ocean. And then there would also be the role played by the imposed depth of the mixed layer itself.

Thank you. We have added in the manuscript:

*Here, a one-dimensional column mixed layer ocean model is employed, without accounting for oceanic advection. Notably, in regions such as the Labrador Sea and along the east coast of Greenland,*
*the prevailing southward ocean currents play a significant role in transporting and dispersing sea ice towards the south. Consequently, neglecting sea ice export associated with ocean currents is likely to result in a positive bias in SIE, particularly over extended lead times.*

5. Figure 4 and spatial biases discussed in (previous) sub-section 3.1: Although I understand the plotting of ice concentration fields for October 6.5 month forecasts: It matches your ability to show similar plots with observation for ice thickness in Figure 5. However it is the April initialized forecast lead with the smallest bias (Figure 1a; Figure 11; are AMSR2 observations

really only available Oct-Apr?), which hides significant system biases, particularly in the Barents and Kara Seas not mentioned explicitly until sub-section 3.4. Some better co-referencing of spatial biases with hemispheric biases and a better coherence with regards to which start dates to concentration on could have been achieved in the text [and I should have taken more careful note in the first revision of this paper].

Thank you. This is a very good point. We added October initializations in Fig. 11 to show the results are similar to the April initializations. AMSR2 observations are available all year along, but not available before 2013 at the time of this study.

We added a discussion on Barents and Kara Seas in Sec.3.2:

*In both the control and CS2_IC experiments, a positive bias in SIT at 6-month lead time is seen in the Barents and Kara Seas, as shown in Fig. 5. This bias is likely attributed to the absence of southward ocean currents in the mixed layer ocean model and the subsequent southward drift of ice, as previously mentioned.*

6. Figure 9. Due to some of the lack of connections made in my previous point the plot of SST biases (Figure 9) lacks cohesion with the rest of the manuscript. The influence (or lack there-of) of oceanic heat transport is obvious from the figure, but then it is hard to then compare this with the sea ice results already shown– or to possibly link the contribution of other system biases (CFSR forcing / depth of mixed layer) to both SST biases and sea ice biases.

Thank you. We agree and have decided to remove the SST bias plot (Fig. 9), but we will retain the discussion of SST.